# Mamba State-Space Models Are Lyapunov-Stable Learners

## Abstract

Mamba (Gu & Dao, 2024a; Dao & Gu, 2024) state-space models (SSMs) were recently shown to outperform state-of-the-art (SOTA) Transformer large language models (LLMs) across various tasks. Despite subsequent widespread adaptation, little work has focused on Mamba LLMs' amenability for fine-tuning frameworks ubiquitously used for Transformer-based LLMs, e.g., mixed-precision fine-tuning (MPFT) and parameter-efficient fine-tuning (PEFT). For the former, it currently remains an open question whether Mamba's recurrent dynamics are robust to small input changes, such as those encountered during MPFT. Using dynamical systems theory (in particular, Lyapunov exponents), we answer this question in the affirmative. We empirically validate this result through several experiments, showing that Mamba SSMs are significantly more stable to changes introduced by mixed-precision than comparable Transformers, even when both MPFT and PEFT are combined. For PEFT, we show how targeting specific memory buffers in Mamba's customized CUDA kernels for low-rank adaptation regularizes SSM parameters, thus providing both parameter efficient learning and computational savings. Finally, with both MPFT and PEFT enabled, we explore the impact of instruction tuning Mamba SSMs for in-context learning (ICL) on *natural language tasks*. While pretrained Mamba and Mamba-2 models only achieve 38% and 82% (respectively) of the ICL improvements of comparable Transformer-based LLMs, we show that instruction tuning allows Mamba models to narrow this gap to 81% and Mamba-2 models to skyrocket over this gap to 132%.

## 1 Introduction

Innovating on previous state-space models (SSMs) (Gu et al., 2022; Dao et al., 2023), Mamba (Gu & Dao, 2024a; Dao & Gu, 2024) has been recently proposed as an accurate, sub-quadratic alternative to Transformer large language models (LLMs). Upon their introduction in Gu & Dao (2024a), Mamba SSMs were shown to greatly outperform comparable attention-based LLMs (Biderman et al., 2023) across a large number of standard natural language benchmarks. Subsequently, pretrained Mamba models have been widely adapted across a large number of data modalities (Liu et al., 2024; Li & Chen, 2024; Quan & Li, 2024; Li et al., 2024), tasks (Xie et al., 2024; Wang et al., 2024a), and architectures (Anthony et al., 2024; Park et al., 2024; Lieber et al., 2024). Despite such widespread adaptation and subsequent research threads (Dao & Gu, 2024; Park et al., 2024; Wang et al., 2024b), little work has been done to understand the amenability of Mamba SSMs for widely used fine-tuning frameworks, such as mixed-precision fine-tuning (MPFT) (Micikevicius et al., 2018) and parameter-efficient fine-tuning (PEFT) (He et al., 2021; Hu et al., 2021).

MPFT and PEFT are arguably two of the most widely utilized techniques for LLM alignment (Tunstall et al., 2023) and customization (VM et al., 2024), and are typically combined to drastically decrease hardware demands needed to fine-tune modern LLMs (Dettmers et al., 2024). However, direct application of MPFT for Mamba SSMs is made difficult due to potential sensitivities of Mamba's state-space dynamics, a common concern for recurrent-based deep models (Pascanu et al., 2013). To combat this, both Huggingface (2024) and Gu & Dao (2024b) suggest full precision (`FP32`) may be required to perform stable training for Mamba models. Thus, it is currently an open question whether Mamba's recurrent dynamics are stable in the presence of small input deviations, such as those introduced in MPFT.

To answer this question, we leverage theory from dynamical systems. Deriving and bounding the Lyapunov exponents for both Mamba and Mamba-2 models, we show that small input changes within the SSM layer of either model do not lead to exponentially deviating outputs. Empirically, we validate this theoretical result; compared to full-precision, deviations due to mixed-precision for Mamba inference are on par with those demonstrated by Transformer LLMs, while deviations due to MPFT are significantly more stable than those of comparable Transformers (Section 4). Furthermore, this trend continues when MPFT and PEFT are combined, where Mamba SSMs again produce significantly smaller deviations compared to comparable Transformer LLMs.

For PEFT, we show that by targeting the large memory buffers exploited by Mamba's highly customized CUDA kernels, LoRA may be used for extremely efficient fine-tuning, while simultaneously regularizing the majority of Mamba's SSM parameters via weight tying. We show that this leads to extremely efficient PEFT, resulting in up to 2.15 times faster training and 65.5% reduced memory compared to the largest evaluated Mamba model without MPFT or PEFT. Furthermore, this allows even the largest (2.8 billion parameter) Mamba LLMs to be fine-tuned on a single GPU with as little as 24GB of onboard memory.

Finally, using both MPFT and PEFT, we complement existing studies (Park et al., 2024; Lee et al., 2024) by exploring the ICL capabilities of instruction-tuned Mamba and Mamba-2 models on *natural language tasks*. In particular, the ICL capabilities of both pretrained Mamba and Mamba-2 models lag behind those of comparable Transformer models from the Pythia suite (Biderman et al., 2023); averaged across five standard natural language benchmarks and foundation model sizes, Mamba and Mamba-2 models only achieve 38% and 82%, respectively, of the performance improvements (relative to zero-shot) of Pythia models. However, after instruction-tuning, Mamba models are able to achieve as much as 81.5% of the average few-shot learning improvement (relative to zero-shot) of comparable Transformers, while Mamba-2 models push this to 132% of the ICL improvements achieved by Pythia models. We note that, similar to Transformer foundation models (Wei et al., 2022), (post) instruction tuning ICL appears as an emergent abilities for Mamba and Mamba-2 SSMs, manifesting for models of size 370 million parameters and greater, while failing to manifest for Mamba and Mamba-2 models of fewer parameter counts.

**Summary of contributions**. Our major contributions are as follows:

- We derive bounds on the Lyapunov exponents of both Mamba and Mamba-2 models' SSM equations. Using these bounds, we theoretically show that small input changes within the SSM layer do not lead to exponentially deviating outputs.

- Empirically, we extensively demonstrate the above theoretical result; across two fine-tuning datasets, two widely used natural language benchmarks, several model sizes, and a large number of MPFT/PEFT configurations, we show that training Mamba LLMs is significantly more stable than comparable Transformer-based LLMs.

- For PEFT, we theoretically show that targeting specific weights for LoRA within Mamba and Mamba-2 SSM layers necessarily leads to weight tying the majority of time-varying parameters. We empirically demonstrate such regularization can improve generalization.

- We complement recent studies by using MPFT and PEFT to understand the ICL capabilities of Mamba/Mamba-2 models evaluated on *natural language tasks*. We show that ICL is an emergent ability of instruction tuned Mamba/Mamba-2 models, and that instruction tuning allows SSMs to perform ICL competitively with comparable Transformer LLMs on natural language tasks.

**Terminology**. We note that herein, when describing a particular foundation model or result, we use the term "Mamba model" to refer to one of the original models released in Gu & Dao (2024a) and "Mamba-2 model" to refer to models released in Gu & Dao (2024a). While there are subtle architectural differences between these two SSMs, they share important similarities which allow our theoretical results to extend to both sets of models. In particular, Mamba and Mamba-2 models share the same state-space equations, support for SSM matrices, and design scheme of storing the majority of SSM parameters in a large memory buffer. Thus, we synonymously use the term `MambaBlock` to refer to the SSM layer of both Mamba and Mamba-2 models.

## 2 MAMBA STATE-SPACE MODELS

For latent-variable dimension $d$ and maximum input sequence length $T$, the `MambaBlock` defines state-space parameters $\mathbf{A}, \mathbf{B}_t, \mathbf{C}_t, \boldsymbol{\Delta}_t \in \mathbb{R}^{d \times d}$ for $t \in \{1, \ldots, T\}$. The matrix $\boldsymbol{\Delta}_t$ controls the discrete step-size. Given an input sequence $\mathbf{u}_1, \ldots, \mathbf{u}_T \in \mathbb{R}^d$, the following linear mapping through latent states $\boldsymbol{x}_1, \ldots, \boldsymbol{x}_T \in \mathbb{R}^d$ is used to produce the output $\mathbf{y}_1, \ldots, \mathbf{y}_T \in \mathbb{R}^d$:

$$\boldsymbol{x}_t = \bar{\mathbf{A}}_t \boldsymbol{x}_{t-1} + \bar{\mathbf{B}}_t \mathbf{u}_t \tag{1}$$

$$\mathbf{y}_t = \bar{\mathbf{C}}_t \boldsymbol{x}_t, \tag{2}$$

where $\bar{\boldsymbol{\Delta}}_t = \mathtt{softplus}(\mathtt{Linear}(\boldsymbol{\Delta}_t)) \in \mathbb{R}^{d \times d}$, $\bar{\mathbf{A}}_t = \exp(\bar{\boldsymbol{\Delta}}_t \mathbf{A})$ and $\bar{\mathbf{B}}_t = \mathbf{A}^{-1}(\bar{\mathbf{A}} - \mathbf{I})\mathbf{B}_t$. In practice, $\mathbf{A}, \mathbf{B}_t, \mathbf{C}_t$ and $\boldsymbol{\Delta}_t$ are diagonal matrices.

### 2.1 STABLE DYNAMICS IN THE MAMBABLOCK

The Mamba foundation models were pretrained in full `FP32` precision. Consequently, official Mamba implementations have cautioned against fine-tuning or training in reduced precision (Gu & Dao, 2024b; Huggingface, 2024), with potential sensitivities of `MambaBlock` recurrent dynamics remaining an open question. We answer the latter using theory from dynamical systems. For Mamba's discrete dynamic system in Equations 1 and 2, define

$$\boldsymbol{x}_t = F_\theta(\boldsymbol{x}_{t-1}, \mathbf{u}_t), \tag{3}$$

where $\theta$ denotes the time-varying parameters described in Section 2. For input sequence $\mathbf{u}_1, \ldots, \mathbf{u}_T$ and initial latent state vector $\boldsymbol{x}_0$, we thus write

$$\boldsymbol{x}_T = F_\theta(F_\theta(\ldots F_\theta(\boldsymbol{x}_0, \mathbf{u}_1))) \coloneqq F_\theta^{T-1}(\boldsymbol{x}_0, \mathbf{u}_1).$$

The rate of divergence between two scalar $\varepsilon$-close inputs to a discrete dynamical system is bounded by the system's maximal Lyapunov exponent $\lambda_{\mathtt{max}}$ (Mikhaeil et al., 2022). Given $\lambda_{\mathtt{max}}$ and two initial values $(\boldsymbol{x}_0, \mathbf{u}_1)$ and $(\boldsymbol{x}_0 + \varepsilon, \mathbf{u}_1 + \varepsilon)$, the maximum deviation between these points grows as (Laffargue et al., 2013; Sayama, 2015):

$$\max |F_\theta^N(\boldsymbol{x}_0, \mathbf{u}_1) - F_\theta^N(\boldsymbol{x}_0 + \varepsilon, \mathbf{u}_1 + \varepsilon)| \in \mathcal{O}(\varepsilon \exp(N\lambda_{\mathtt{max}})).$$

Thus, when $\lambda_{\mathtt{max}} > 0$, nearby trajectories exponentially separate and, when $\lambda_{\mathtt{max}} \leqslant 0$, nearby trajectories ultimately converge to the same fixed point or periodic cycles.

The maximal Lyapunov exponent is defined as

$$\lambda_{\mathtt{max}} \coloneqq \lim_{T \to \infty} \frac{1}{T} \log \left\| \prod_{t=0}^{T} \frac{\partial \boldsymbol{x}_t}{\partial \boldsymbol{x}_{t-1}} \right\|_2,$$

where $\|\|_2$ denotes the spectral norm for matrices. For an arbitrary `MambaBlock`, we prove the following:

**Theorem 1.** *Let $(\boldsymbol{x}_{t-1}, \mathbf{u}_t)$ be the latent state and input at an arbitrary time $t \in \{1, \ldots, T\}$ within a `MambaBlock`. Then small changes $(\boldsymbol{x}_{t-1} + \varepsilon, \mathbf{u}_t + \varepsilon)$ produce deviations which are exponentially non-increasing over discrete-time. That is, $\max |F_\theta^N(\boldsymbol{x}_{t-1}, \mathbf{u}_t) - F_\theta^N(\boldsymbol{x}_{t-1} + \varepsilon, \mathbf{u}_t + \varepsilon)| \in \mathcal{O}(\varepsilon \exp(N\zeta))$, for some scalar $\zeta \leqslant 0$.*

The proof of Theorem 1 is available in Appendix B, where the maximal Lyapunov exponent for an arbitrary `MambaBlock` is first proven to be non-positive. The main result subsequently follows.

Thus, the latent states of Mamba and Mamba-2 models are stable under small input changes. However, variables $\mathbf{y}_1, \ldots, \mathbf{y}_T$ are the primary outputs for such models, particularly for LLM applications. We next show that, given Theorem 1, Mamba and Mamba-2 output variables are also stable.

**Theorem 2.** *Assume $(\boldsymbol{x}_{t-1} + \varepsilon, \mathbf{u}_t + \varepsilon)$ produce deviations which are exponentially non-increasing over discrete-time. Then small changes to the output $\mathbf{y}_t$ are also exponentially non-increasing over discrete time.*

The proof of Theorem 2 is available in Appendix C. Thus, by Theorems 1 and 2, the latent and output states of both Mamba and Mamba-2 models are stable to changes encountered during recurrency.

### 2.1.1 CONSEQUENCES FOR AUTOMATIC MIXED-PRECISION

During a forward pass, automatic mixed-precision (AMP) saves time and memory by computing forward activations in half-precision (`FP16` or `BF16`). During a backward pass, AMP computes gradients in half-precision and up-casts to full-precision prior to updating. In contrast to full-precision fine-tuning, MPFT within the `MambaBlock` thus results in small differences to the inputs $\mathbf{u}_1, \ldots, \mathbf{u}_T$ (which are passed through a `Swish`), $\bar{\boldsymbol{\Delta}}_t$ (which is passed through a `softplus`), and the gradients calculated during training.

For a discrete dynamical system with $\lambda_{\max} > 0$, changes due to AMP compound after repeated expansion of the recurrent state, thus leading to exponential deviations between quantities calculated using mixed- versus full-precision. We note that Transformers are not recurrent, and thus not susceptible to such issues. Yet, just as differences introduced by quantization/mixed-precision produce output differences in Transformer results, differences are expected in Mamba results using different precision strategies. However, by Theorem 1, such differences do not exponentially compound over discrete-time within the `MambaBlock`.

### 2.2 HARDWARE-AWARE OPTIMIZATIONS AND PEFT

As matrices $\mathbf{B}_t, \mathbf{C}_t$ and $\boldsymbol{\Delta}_t$ are time-varying, S4 optimizations via the SSM convolution kernel (Dao et al., 2023) are no longer applicable. However, by diagonality, each dimension may be computed in parallel. Furthermore, the recurrence along every dimension is a prefix sum (also called a *scan*), which is highly parallelizable (Blelloch, 1990). Gu & Dao (2024a) thus capitalizes on this through extensively customized CUDA kernels wherein the majority of temporal variables are carefully laid out in a large buffer of GPU memory and manipulated. Instantiated as a `PyTorch` linear layer's weight matrix, this memory buffer $\mathbf{W} \in \mathbb{R}^{T \times 3d}$ is used to store and access the diagonal elements of $\mathbf{B}_t, \mathbf{C}_t$ and $\boldsymbol{\Delta}_t$ for all $t \in \{1, \ldots, T\}$, such that

$$\mathbf{W}[t-1, :d] = \texttt{diag}(\boldsymbol{\Delta}_t), \mathbf{W}[t-1, d:2d] = \texttt{diag}(\mathbf{B}_t), \mathbf{W}[t-1, 2d:3d] = \texttt{diag}(\mathbf{C}_t), \tag{4}$$

where $\mathbf{W}[0, :d] = \texttt{diag}(\boldsymbol{\Delta}_1), \mathbf{W}[n-1, d:2d] = \texttt{diag}(\mathbf{B}_T)$, and so on. The customized Mamba prefix scan kernel heavily relies on this memory layout to optimize the access pattern of $\mathbf{W}$ in Equations 1 and 2.

Similarly, Mamba-2 stores diagonal elements of $\mathbf{B}_t, \mathbf{C}_t$ and $\boldsymbol{\Delta}_t$ in a large memory buffer $\mathbf{W}$. However, rather than utilizing the underlying recurrence to directly compute the hidden state and output at each time-step, Dao & Gu (2024) consider the matrix resulting from unrolling Equation 2 across all $t$. Mamba-2 models thus leverage the structure of the ensuing semiseparable matrix to calculate $\boldsymbol{x}_1, \ldots, \boldsymbol{x}_T$ and $\mathbf{y}_1, \ldots, \mathbf{y}_T$ using tensor contractions, which are highly optimized on modern hardware accelerators.

When fine-tuning using LoRA, low-rank matrices are used to adapt the frozen weight matrices of targeted linear layers. The importance of $\mathbf{W}$ for both Mamba and Mamba-2 models makes it a primary candidate for LoRA adaptation. In such cases, selecting $\mathbf{W}$ for LoRA adaptation results in the following:

**Theorem 3.** *Consider the weight matrix* $\mathbf{W}$ *of a* `MambaBlock` *from Equation 4. Targeting* $\mathbf{W}$ *for LoRA during fine-tuning ties adaptation weights across* $\mathbf{B}_t, \mathbf{C}_t$ *and* $\boldsymbol{\Delta}_t$.

The proof of Theorem 3 is available in Appendix D. The specific affects of both targeting $\mathbf{W}$ for fine-tuning and Theorem 3's impact on generalization are ablated in Section 4.2.

## 3 RELATED WORK

Recent work has sought to understand how to efficiently increase Mamba's hidden-state dimension by restructuring SSM operations using tensor contractions (Dao & Gu, 2024), leading to Mamba-2 models. A separate line of work has sought to understand the in-context learning (ICL) capabilities of Mamba LLMs when trained from scratch for specific tasks (Park et al., 2024; Lee et al., 2024). Another line of recent work has sought to understand how hybrid Mamba-Transformer models may be directly distilled from Transformer models (Wang et al., 2024b). However, to the best of our

knowledge, no existing works have either theoretically explored the effects small input changes (e.g., due to mixed-precision) have on Mamba's recurrent dynamics, empirically explored such effects downstream impact on fine-tuning and inference, or sought to understand the effects of LoRA adaptation on modules within the `MambaBlock`.

Lyapunov exponents have previously been considered for classic RNN structures (e.g., vanilla RNNs, LSTMs, GRUs, PLRNNs, etc.) (Mikhaeil et al., 2022; Vogt et al., 2022), to determine when such models exhibit chaotic dynamics and the impact on the exploding/vanishing gradient phenomena[1]. For more recent S4 neural models, (Goel et al., 2022) used Hurwitz matrices to characterize the numerical stability of linear time-invariant (LTI) S4 models. However, such analysis is not applicable to time-varying models, such as Mamba, nor does it characterize the effects of sensitive dependence on initial conditions (e.g., divergence of two $\varepsilon$ close inputs). To the best of our knowledge, no previous works have used Lyapunov exponents to explore the effects of mixed-precision on recurrent neural models or Mamba architectures.

As previously noted, recent works (Park et al., 2024; Lee et al., 2024) have studied Mamba's ability to perform ICL by training Mamba models for specific tasks. Such tasks include logistic regression, decision trees, and learning other simple function classes, following the work of Garg et al. (2022). We emphasize that, in this set up, relatively small Mamba models–33 million and 90 million parameters for Lee et al. (2024) and Park et al. (2024), respectively–are trained from scratch for every evaluated task. Indeed, Park et al. (2024) notes that subsequent work is necessary to understand Mamba's ICL capabilities for language modeling using standard natural language benchmarks, as well as for larger model sizes. Thus, our study of both the pretrained and instruction tuned ICL capabilities of Mamba/Mamba-2 LLMs for natural language tasks are complimentary to previous works.

## 4 EXPERIMENTS

To demonstrate the implications of Theorem 2, we explore the performance difference between running inference with full-precision pretrained weights and using mixed-precision (`FP16` and `BF16`) weights. **Model performance is measured as percent accuracy** using the MMLU dataset (Hendrycks et al., 2020). The difference in model performance is reported as the mean *divergence* (i.e., absolute difference) between the original full-precision and respective mixed-precision model, averaged over {0, 1, 3, 5}-shot percent accuracy. Thus, **a divergence greater than one denotes an average difference greater than one entire percentage of accuracy.**

Mamba pretrained checkpoints are compared to pretrained Transformer models of similar parameter counts and no more than ~300B total pretraining tokens (Pythia (Biderman et al., 2023), OLMo (Groeneveld et al., 2024) 336B-token checkpoint, and Phi 1.5 (Li et al., 2023)). We note that **Pythia and Mamba models were both pretrained using the same corpus** (Gao et al., 2020), **allowing the fairest comparison between SSMs and Transformers**. To limit extraneous numerical effects within experiments (e.g., due to parameter aggregation across multiple GPUs), all models were run using a single GPU (Nvidia A10G, 24 GB total memory). All models were evaluated using the LM evaluation harness from Eleuther AI (Gao et al., 2023). Further experimental details are available in Appendix E. The results are available in Table 1.

Table 1: Mean full-precision (`FP32`) divergence in MMLU performance for mixed-precision inference. Divergence is averaged over {0, 1, 3, 5}-shot performance. Pretrained checkpoints are used for Mamba (`M`), Pythia (`P`), OLMo (Groeneveld et al., 2024), and Phi-1.5 (Li et al., 2023) (`Phi`) models.

| Model | M | P | M | P | M | P | OLMo | M | P | Phi | M | P |
|-------|-----|-----|-----|-----|-----|----|------|-----|-----|-----|-----|-----|
| Size | 130M | 160M | 370M | 410M | 790M | 1B | | 1.4B | 1.5B | | 2.8B | |
| `FP16` $\mu$ | 0.03 | 0.35 | 0.05 | 0.06 | 0.21 | 0.05 | 0.04 | 0.04 | 0.07 | 0.03 | 0.15 | 0.12 |
| `BF16` $\mu$ | 0.05 | 1.45 | 0.20 | 0.20 | 0.66 | 0.16 | 0.13 | 0.31 | 0.13 | 1.05 | 1.17 | 0.11 |

---

[1]We note that this continues a long line of research exploring RNNs sensitivity to initial conditions and their subsequent ability to produce chaotic output (Ribeiro et al., 2020; Laurent & von Brecht, 2017; Bertschinger & Natschläger, 2004; Bertschinger et al., 2004), although previous work did not leverage Lyapunov exponents.

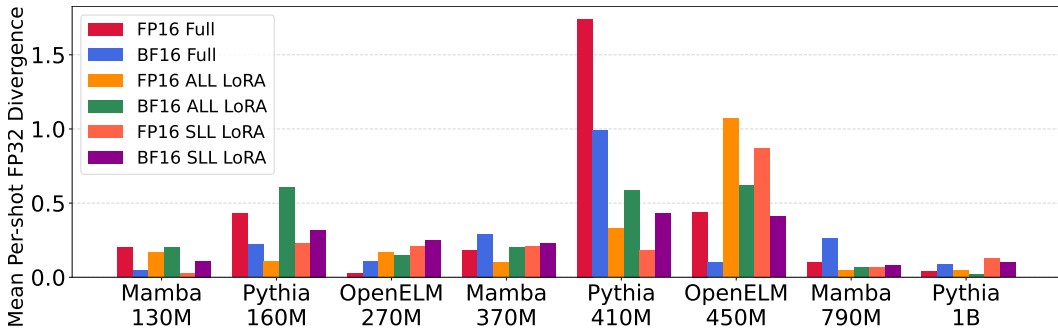

Figure 1: Mean full-precision (`FP32`) divergence in MMLU performance for Mamba, Pythia, and OpenELM models. Models are fine-tuned over the `Alpaca` dataset using different combinations of MPFT and PEFT. Full fine-tuning (i.e., no PEFT adapters) is denoted as `Full`.

From Table 1, inferencing in Pythia using `FP16` and `BF16` result in an average 0.13 and 0.41 full-precision divergence, respectively. Mamba displays similar averages in comparison: inferencing in Mamba using `FP16` and `BF16` result in an average 0.10 and 0.48 divergence, respectively. Interestingly, both SSM and Transformer architectures exhibit *large divergence spikes*–i.e., mean divergence greater than a percentage point–when using `BF16`, which occurs once for Mamba and Phi 1.5 models and twice for Pythia models. Due to space constraints, Mamba-2 results for the same experiment are included in Appendix Table F. We note that, for comparable model sizes and mixed-precision, Mamba-2 models follow identical divergence trends as respective Mamba models.

In the following, we show that the observed large divergence spikes may be mitigated for Mamba SSMs by combining mixed-precision with parameter-efficient adapters during fine-tuning.

**Non-divergent Mamba fine-tuning.** We next explore the implications of Theorem 1 on fine-tuning, wherein mixed-precision is especially critical; MPFT combined with PEFT adapters have been shown to drastically reduce Transformer fine-tuning times (Dettmers et al., 2024). We are thus interested in the divergence between Mamba models fully fine-tuned (i.e., no adapters, all model weights are trained) in full-precision and models fine-tuned using mixed-precision and/or PEFT adapters. We focus on utilizing LoRA (Hu et al., 2021), which is arguably the most widely used PEFT framework for LLMs.

Using the `Alpaca` dataset (Taori et al., 2023), `Mamba 160M`, `410M`, and `790M` models are fine-tuned for three epochs with a maximum sequence length of 512. We denote the targeting of all linear layers (ALL) for LoRA as *ALL LoRA*, the targeting of a subset of linear layers (SLL) for LoRA as *SLL LoRA*, and no adapters as *Full* (i.e., full fine-tuning). Both `ALL` and `SLL` LoRA adapt the large memory buffer described in Theorem 3.

Each fine-tuning run occurred on a single A10G GPU. To further limit extraneous numerical effects, the same batch size is used for all `FP32`, `FP16`, and `BF16` experiments for a given model size. While this leads to hardware underutilization (i.e., non-saturated GPU memory for mixed-precision and LoRA experiments), this is necessary to guarantee no divergence is due to differences in parameter update schedules. For comparison, two Transformer-based LLM families of similar parameter counts are fine-tuned using the same experimental setup: `Pythia` (sizes `160M`, `410M`, and `1B`) and `OpenELM` (Mehta et al., 2024) (sizes `270M` and `450M`). The training recipe for all models was adapted from (Tunstall et al., 2023), with the `AdamW_torch` optimizer and a `cosine annealing` schedule. Further experimental details are available in Appendix E.

For each Mamba, Pythia, and OpenELM model, Figure 4 shows the mean divergence calculated between the respective `FP32 Full` and mixed-precision `ALL`/`SLL LoRA` fine-tuned models, averaged over {0, 1, 3, 5}-shot MMLU accuracy.

Across mixed-precisions and adapter settings, Mamba displays smaller divergences than both Pythia and OpenELM models. E.g., **for `FP16`, Mamba demonstrates an average divergence of 0.1, compared to 0.14 for Pythia and 0.54 for OpenELM**. Similarly, for **`BF16`, Mamba demonstrates an average divergence of 0.18, compared to 0.28 for Pythia and 0.33 for OpenELM**. Importantly,

Mamba models do not exhibit large deviation spikes after fine-tuning, in contrast to both Pythia and OpenELM models. Further experiments with additional fine-tuning and benchmark datasets are available in Appendix I.

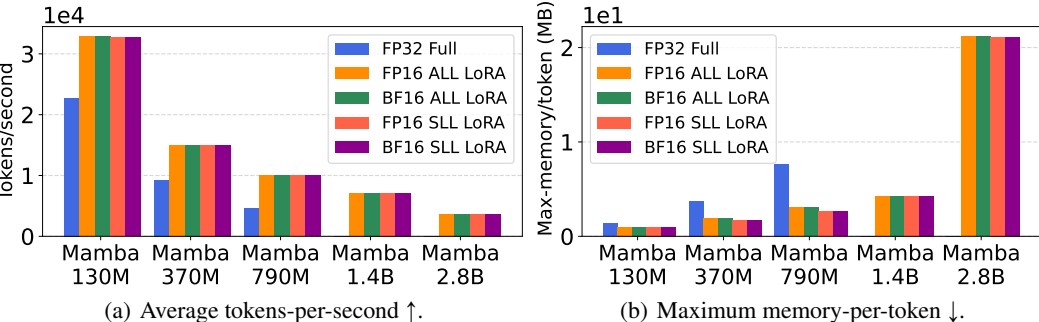

(a) Average tokens-per-second ↑.

(b) Maximum memory-per-token ↓.

Figure 2: Timing and memory usage calculated Mamba model-sizes and PEFT combinations. Each model was trained using the Alpaca dataset dataset for three epochs and maximum sequence length 512. For each PEFT combination, the batch size was tuned to maximize GPU occupancy. Full fine-tuning exceeds available GPU memory (24 GB) for models greater than 790 million parameters.

**Hardware throughput and memory-utilization improvements**. With stable dynamics and observed divergences smaller than comparable Transformers, we show that MPFT and PEFT may be used to significantly increase GPU-training throughput for Mamba SSMs. To demonstrate such improvements, we utilize the previous fine-tuning settings for the Alpaca dataset. However, we now adjust the batch size to maximize throughput per MPFT and PEFT configuration.

For each MPFT and PEFT configuration, the *average tokens-per-second* (ATPS) is calculated as the total tokens used for fine-tuning divided by total training time, and the *maximum memory-per-token* (MMPT) is calculated as the maximum GPU memory utilization incurred (over the entire fine-tuning run) divided by the total number of tokens in each mini-batch. Results are plotted in Figure 4.

Both throughput and memory utilization improve as the number of Mamba parameters increases in Figure 4. **Compared to the full-precision full fine-tuning of `Mamba 790M`** (the largest model supported by an `A10G`'s memory capacity), evaluated **MPFT and PEFT combinations result in an average 2.15 times more training tokens-per-second while reducing per-token memory utilization by an average 62.7%**. Across all model sizes, evaluated MPFT and PEFT combinations result in an average 1.74 times more training tokens-per-second while reducing per-token memory utilization by an average 47.2% compared to respective full-precision fine-tuned runs. Furthermore, while full fine-tuning is no longer possible on a single `A10G` for Mamba models greater than 790 million parameters, MPFT and PEFT allow training Mamba models up to 2.8 billion parameters on GPUs with as little as 24 GB onboard memory.

### 4.1 INSTRUCTION TUNING IMPACT ON MAMBA ICL FOR NATURAL LANGUAGE TASKS

Using both MPFT and PEFT, we next explore how instruction tuning affects Mamba and Mamba-2 ICL performance on natural language tasks. All Mamba and Mamba-2 pretrained models are instruction fine-tuned using `ALL LoRA` and the OpenHermes dataset (Teknium, 2024) (which consists of 242,000 supervised samples). We use the training recipe of (Tunstall et al., 2023), which includes `BF16` utilization.

Zero and few-shot performance is evaluated using five standard natural language benchmarks: HellaSwag (Zellers et al., 2019), PIQA (Bisk et al., 2020), Arc-E (Clark et al., 2018), Arc-C (Clark et al., 2018), and WinoGrande (Sakaguchi et al., 2021). ICL performance is reported as the *average improvement percentage* of $\{1, 3, 5\}$-*shot* versus 0-*shot* (AIPSS). For comparison, Pythia pretrained models are instruction fine-tuned using the same training recipe and `ALL LoRA` (i.e., all Pythia linear layers are adapted).

Figure 3 displays AIPSS for pretrained and instruction fine-tuned Mamba and Pythia models. As previously noted, pretrained Mamba models do not display similar ICL ability as comparable

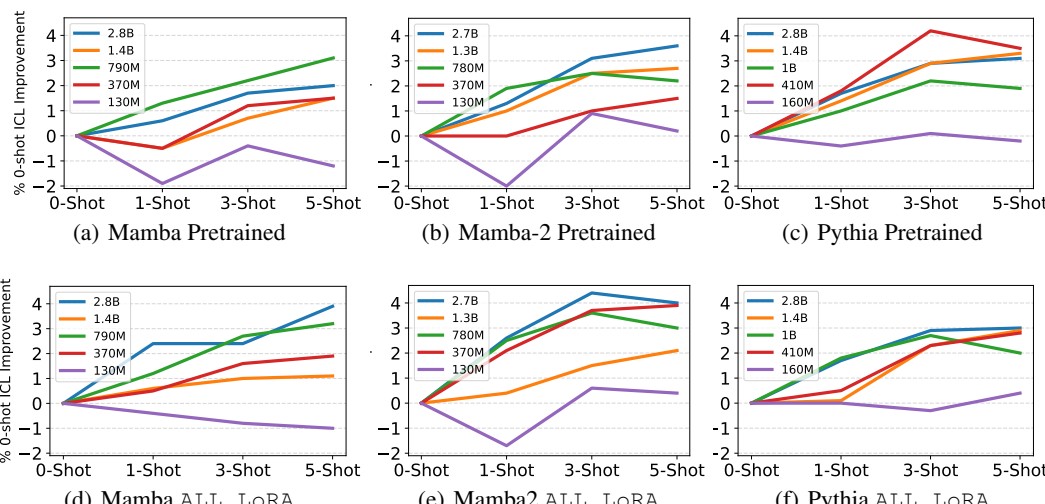

Figure 3: Instruction tuning narrows the ICL gap between Mamba and Pythia, and creates a gap from Pythia to Mamba-2 models. `ALL LoRA` models were instruction tuned on the OpenHermes (Teknium, 2024) dataset for one epoch. Performance is reported as the average improvement percentage of {1, 3, 5}-shot versus 0-shot over five standard natural language benchmarks.

Pythia models on the evaluated standard NLP benchmarks. In particular, `Mamba 2.8B`, the largest pretrained Mamba model, displays inconsistent zero-shot improvements as the number of shots increase. While pretrained Mamba-2 models display significantly better ICL ability than Mamba models, Mamba-2 models smaller than 780 million parameters struggle.

However, after instruction tuning, all Mamba models larger than `Mamba 130M` consistently improve in ICL performance as the number of shots increase. Similarly, the majority of Mamba-2 models larger than `Mamba 130M` greatly improve in ICL performance. Thus, while pretrained Mamba and Mamba-2 models are only capable of 38% and 82% (respectively) of the AIPSS compared to similar pretrained Pythia models, instruction tuned Mamba and Mamba-2 models are capable of 81.5% and 133% of the AIPSS relative to similarly fine-tuned Pythia models. We note that a significant difference between Mamba and Mamba-2 models is the larger (by a factor of four) latent dimension.

**ICL as an emergent ability of Mamba SSMs**. We next study the emergent behavior (as a function of model size) of Mamba/Mamba-2 SSMs' ICL abilities on natural language tasks by comparing to a larger number of Transformer-based LLMs of varying sizes. We compare to `OpenELM` (Mehta et al., 2024) (sizes `270M`, `450M`, and `1.1B`), `TinyLlama 1.1B` (Zhang et al., 2024), and `OLMo 1.2B` (Groeneveld et al., 2024). To limit the emergent effects on both parameter size and pretraining token counts, we did not evaluate models greater than 2.8 billion parameters and chose open-source checkpoints as close as possible to the 300 billion total pretraining tokens used for Mamba, Mamba-2, and Pythia models. Thus, pretraining token counts for OpenELM, TinyLlama, and OLMo models were 429 billion, 503 billion, and 336 billion, respectively. We note this potentially biases ICL performance in favor of the newly evaluated Transformer-based LLMs, and that direct comparisons between Mamba, Mamba-2, and Pythia are the most fair (as these three classes of models were all pretrained on the same dataset for the same number of total pretraining tokens).

We repeat the experiments from Figure 3, where we evaluate the pretrained and instruction tuned ICL capabilities of all models. To understand the critical role of parameter counts, we group all models into two classes: LLMs containing 450 million parameters or less, and LLMs containing greater than 450 million parameters. ICL performance measured by AIPSS is displayed in Figure 4.

From Figure 4, it is clear that pretrained SSMs and Transformers of parameter counts 270 million and less display slight or detrimental ICL abilities (i.e., few-shot performance is worse than zero-shot). For models of greater than 450 million parameters, the majority of SSMs and Transformers display positive ICL abilities, with `Mamba 1.4B` being an outlier in terms of poor performance. With the

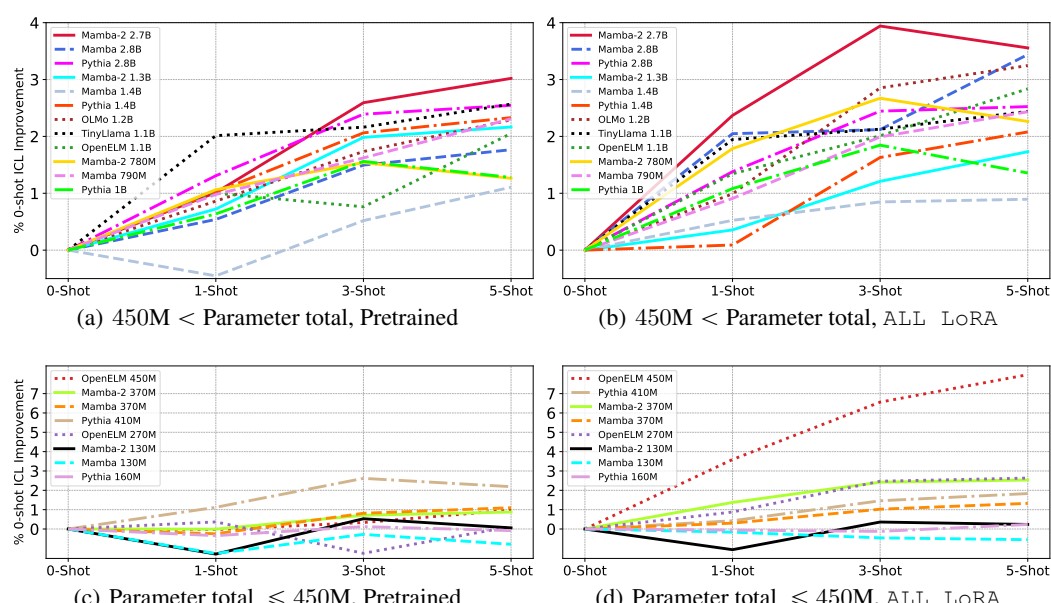

Figure 4: Instruction tuning improves Mamba-2 ICL performance past Transformer LLMs. `ALL LoRA` models were instruction fine-tuned on the OpenHermes dataset for one epoch. Performance is reported as the average improvement percentage of {1, 3, 5}-shot versus 0-shot over five standard natural language benchmarks: HellaSwag, PIQA, Arc-E, Arc-C, and WinoGrande.

exception of TinyLlama at 1-shot performance and `Mamba-2 2.7B` for 3- and 5-shot performance, the majority of other pretrained models cluster together.

Instruction tuning greatly smooths ICL performance across both parameter classes. While instruction tuned SSMs and Transformers of 160 million parameters or fewer continue to display slight or detrimental ICL abilities, all parameters of 270 million and greater show positive ICL abilities. The instruction tuned `OpenELM 450M` model displays particularly impressive ICL abilities, but it is difficult to determine whether it is strictly due to architecture and/or pretraining recipe, or partially due to 143% more total pretraining tokens than Mamba/Mamba-2 and Pythia models. For instruction tuned models of greater than 450 million parameters, all SSMs and Transformers show positive ICL abilities, with `Mamba-2 2.7B` greatly outperforming all other models (both SSM and Transformer) in this class.

Thus, in terms of ICL as a function of SSM model size, while no clear trend presents itself for pretrained models, ICL appears to emerge for instruction tuned Mamba and Mamba-2 SSMs of size 370 million and greater.

## 4.2 PEFT-LAYER EFFECTS ON MAMBA-2 ZERO-SHOT PERFORMANCE

As described in Section 2.2, both Mamba and Mamba-2 store the majority of time-varying SSM parameters in a large memory buffer, denoted as $\mathbf{W}$. For PEFT via LoRA, we ablate the impact of targetting $\mathbf{W}$ and, thus, demonstrate the impact of Theorem 3, where time-varying variables accessed through $\mathbf{W}$ were shown to be regularized through weight tying.

Firstly, using the dataset and settings from Section 4.1, we instruction tune Mamba-2 models using LoRA by both adapting $\mathbf{W}$ and adapting all linear layers other than $\mathbf{W}$. Mamba-2 `MambaBlocks` contain two linear layers, thus the former targets only the $\mathbf{W}$ linear layer whereas the latter targets the other linear layer in each block. As shown in Table 2, PEFT with only $\mathbf{W}$ targetted within the `MambaBlock` near uniformly results in better performance than only targeting other layers, with the former outperforming the latter on 32 out of 35 natural language tasks. This makes intuitive sense since $\mathbf{W}$ represents the majority of Mamba's time-varying parameters (as previously mentioned). Thus, while previous works Gu & Dao (2024a); Dao & Gu (2024) have displayed the importance

of Mamba's time-varying parameters for pretraining performance, this verifies the importance of Mamba's time-varying parameters for instruction tuning performance.

Table 2: Zero-shot performance for instruction tuned Mamba-2 models where: ✓ denotes the large memory buffer **W** containing the majority of temporal variables (described in Section 2.2) is targeted for LoRA adaptation, and ✗ denotes **W** is not adapted. The top-performance for each task per model is highlighted in bold .

| Model | W targeted? | LAMBADA ppl ↓ | LAMBADA acc ↑ | HellaSwag acc ↑ | PIQA acc ↑ | Arc-E acc ↑ | Arc-C acc ↑ | WinoGrande acc ↑ |
|---|---|---|---|---|---|---|---|---|
| Mamba-2 | ✓ | **15.37** | **45.16** | **35.41** | **65.02** | **47.94** | **24.91** | 52.17 |
| 130M | ✗ | 16.95 | 43.30 | 35.24 | 64.85 | 47.47 | 24.06 | 52.17 |
| Mamba-2 | ✓ | **8.03** | **54.16** | **46.87** | 69.53 | 53.49 | **27.73** | **57.14** |
| 370M | ✗ | 8.50 | 53.74 | 46.59 | **70.67** | **54.80** | 26.79 | 55.64 |
| Mamba-2 | ✓ | **5.79** | 61.61 | **55.14** | 71.98 | **61.11** | **29.18** | 60.09 |
| 780M | ✗ | 5.86 | **61.63** | 54.94 | **72.03** | 60.98 | 28.41 | **60.22** |
| Mamba-2 | ✓ | **4.54** | **65.73** | **60.88** | **73.67** | **66.16** | **34.56** | **61.80** |
| 1.3B | ✗ | 5.05 | 65.44 | 59.84 | 73.39 | 64.10 | 33.11 | 61.09 |
| Mamba-2 | ✓ | **4.05** | **69.63** | **66.73** | **76.50** | **69.95** | **36.77** | **64.56** |
| 2.7B | ✗ | 4.10 | 69.61 | 66.60 | 76.39 | 69.53 | 36.26 | 63.93 |

Next, we present a counterintuitive experiment to demonstrate the impact of Theorem 3. As before, we instruction tune Mamba-2 models using LoRA, focusing on the two largest model sizes. This time, however, we target *all linear layers* within Mamba-2's `MambaBlock` and contrast this with targeting *only* **W** within the `MambaBlock`. Displayed in Appendix G Table 7, we can see that adapting more SSM parameters does not necessarily lead to improved performance across the board. Rather, adapting only **W** for Mamba-2 outperforms adapting all linear layers on eight of the 14 natural language tasks.

The result in Table 7 thus presents two cases: (a) *regularization*: only adapting parameters which lead to weight tying (by Theorem 3) and (b) *increased learning capacity*: adapting more parameters, at the cost of learning unregularized parameters. Through regularization via weight tying (Press & Wolf, 2017), (a) leads to good generalization. In contrast, while adapting more parameters, (b) includes unregularized variables, leading to positive improvements on only a minority of tasks when compared to the fine-tuning of fewer, but regularized, parameters. We note that adapting all linear layers is often performed when fine-tuning attention-based LLMs using LoRA Dettmers et al. (2024). Thus, a similar regularization-vs-capacity tradeoff for Transformers LLMs may present itself by carefully studying LoRA's affects when targeting specific linear layers in attention-based architectures.

## 5 DISCUSSION AND FUTURE DIRECTIONS

Using dynamical systems theory, we've shown that the recurrent dynamics of Mamba SSMs are robust to small input perturbations. We've extensively confirmed this result, showing that: a) Mamba inference differences due to mixed-precision align with Transformers, (b) Mamba fine-tuning is significantly more robust to changes due to mixed-precision and PEFT than Transformers, and (c) combining MPFT and PEFT can more than halve training time and nearly triple memory efficiency for Mamba models. Using both MPFT and PEFT, we've shown that instruction tuning Mamba and Mamba-2 SSMs greatly narrows the pretraining ICL gap on natural language tasks relative to comparable Transformer LLMs. In particular, this allows Mamba-2 SSMs to greatly outperform the ICL abilities of a large number of instruction tuned, attention-based LLMs. Furthermore, complimentary to recent studies, we've shown that ICL for natural language tasks can be characterized as an emergent ability of Mamba and Mamba-2 models of 370 millions parameters or greater.

There are several avenues for future work. In particular, adapting Mamba's CUDA kernels to support more aggressive low-precision PEFT methods (Dettmers et al., 2024) would further decrease the hardware needed to train Mamba models, while providing additional speedups and testing the limits of the derived stability results. Furthermore, our theoretical contributions open the door for follow up studies, both in terms of extending our stability results to more general error (and adversarial) robustness results, as well as deriving new SSM-specific LoRA schemes for regularized learning.

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

## A BACKGROUND

**Mixed-precision and parameter-efficient fine-tuning**. In the current era of extremely large foundation models, both MPFT and PEFT have become ubiquitous tools for the rapid adaptation of Transformer-based LLMs towards specific applications. PEFT using adapters (He et al., 2021) allows a large pretrained model to be efficiently adapted for a particular downstream task by freezing the full model and training only a small number of extra parameters. Arguably the most widely used such PEFT method is LoRA (Hu et al., 2021), which injects trainable low-rank matrices into Transformer layers to approximate weight updates.

To further decrease the computational demands necessary for LLM fine-tuning and inference, MPFT via mixed-precision (i.e., `FP16` or `BF16`) (Kalamkar et al., 2019; Micikevicius et al., 2018) and quantized low-precision (Dettmers et al., 2024) have proven effective strategies to reduce GPU memory and runtime requirements without deleterious effects on downstream performance (Dettmers et al., 2024; Wu et al., 2020). Additionally, mixed-precision approaches have paved the way for hardware-aware optimizations within the self-attention module (Dao et al., 2022), greatly mitigating the quadratic complexity of Transformer LLMs. Together, PEFT and MPFT have created a rich ecosystem with which varying combinations of these approaches may be used to meet the computational constraints of a given training system. We note that post-fine-tuning quantization approaches (Frantar et al., 2023) may be further used to decrease Transformer LLM computational demands, but such approaches are not considered in this work.

**State-space Models**. *Structured state-space sequence* (S4) models (Gu et al., 2022; Fu et al., 2023) are SSMs which leverage linear time-invariant (LTI) systems to combine the computational advantages of Transformers–i.e., highly parallelizable training–and recurrent neural networks (RNNs)– i.e., subquadratic autoregressive inference using recurrency. Within the S4 layer, an input signal is discretized and LTI parameters representing the input's latent dynamics are learned. Owing to the S4 block's latent dynamics being LTI, the S4 block's output may be thus compactly represented as

a single convolution between the input and an *SSM convolution kernel* (a matrix whose entries are products of LTI learnable parameters resulting from unrolling the state-space equations). However, despite hardware efficiency and long-dependency-modeling improvements, LTI-based S4 models remained inferior to Transformers of comparable parameter-sizes for natural language tasks, even when augmenting S4 layers with attention-layers for hybrid architectures (Gu & Dao, 2024a).

Innovating on these previous S4 approaches, Mamba utilizes time-*varying* parameters to model latent dynamics, thus broadening the ability to capture nuanced changes evolving in discrete-time. Without LTI dynamics, however, the input-output representation via the SSM convolution kernel is no longer applicable, thus voiding previous hardware-aware S4 optimizations (Fu et al., 2023). To enable hardware efficiency with time-varying SSM parameters, (Gu & Dao, 2024a) thus introduced extensively customized CUDA kernels which implement highly parallelized prefix sums to compute recurrent states. Subsequently, Dao & Gu (2024) considered the unrolled state-space equations and leveraged tensor contractions (i.e., einsum notation (Rogozhnikov, 2022)) to efficiently calculate Mamba variables. The resulting Mamba-2 foundation models contained significantly larger latent-variable dimensions than the Mamba models of (Gu & Dao, 2024a), while maintaining efficiency on modern GPU accelerators.

**In-context learning**. ICL provides an adaptable alternative to fine-tuning. Rather than fine-tune the LLM directly, ICL augments a prompt with $n$ relevant examples (called *shots*) preceding the query of interest. Given sufficiently large models and pretraining data (Brown et al., 2020; Wei et al., 2022), Transformer LLMs have proven adept at learning new concepts on the fly provided such few-shot prompting. However, it is worth noting that ICL inference time increases dramatically as the number of shots grows (due to self-attention's quadratic complexity) and PEFT (when possible) is known to produce more accurate downstream learning results (Brown et al., 2020; Liu et al., 2022).

## B    MAMBA STABLE DYNAMICS PROOF

Recall the state-space parameters and equations for the `MambaBlock`; $\mathbf{A}, \mathbf{B}_t, \mathbf{C}_t, \boldsymbol{\Delta}_t \in \mathbb{R}^{d \times d}$ for $t \in \{1, \ldots, T\} = [T]$. Given an input sequence $\mathbf{u}_1, \ldots, \mathbf{u}_T \in \mathbb{R}^d$, the following linear mapping through latent states $\boldsymbol{x}_1, \ldots, \boldsymbol{x}_T \in \mathbb{R}^d$ is used to produce the output $\mathbf{y}_1, \ldots, \mathbf{y}_T \in \mathbb{R}^d$:

$$\boldsymbol{x}_t = \bar{\mathbf{A}}_t \boldsymbol{x}_{t-1} + \bar{\mathbf{B}}_t \mathbf{u}_t \tag{5}$$
$$\mathbf{y}_t = \bar{\mathbf{C}}_t \boldsymbol{x}_t,$$

where $\bar{\boldsymbol{\Delta}}_t = \texttt{softplus}(\texttt{Linear}(\boldsymbol{\Delta}_t)) \in \mathbb{R}_{\geqslant 0}^{d \times d}$, $\bar{\mathbf{A}}_t = \exp(\bar{\boldsymbol{\Delta}}_t \mathbf{A})$, $\bar{\mathbf{B}}_t = \mathbf{A}^{-1}(\bar{\mathbf{A}} - \mathbf{I})\mathbf{B}_t$, and $\mathbb{R}_{\geqslant 0}$ is the set of non-negative real numbers. In practice, $\mathbf{A}, \mathbf{B}_t, \mathbf{C}_t$ and $\boldsymbol{\Delta}_t$ are diagonal matrices.

Furthermore, recall the following definitions:

$$\boldsymbol{x}_t = F_\theta(\boldsymbol{x}_{t-1}, \mathbf{u}_t)$$

where $\theta$ denotes the aforementioned time-varying parameters. For input sequence $\mathbf{u}_t, \ldots, \mathbf{u}_T$ and initial latent state value $\boldsymbol{x}_0$, we thus write

$$\boldsymbol{x}_T = F_\theta(F_\theta(\ldots F_\theta(\boldsymbol{x}_0, \mathbf{u}_1))) \coloneqq F_\theta^{T-1}(\boldsymbol{x}_0, \mathbf{u}_1).$$

We first prove that, given two scalar $\varepsilon$-close inputs to a `MambaBlock`, their deviations do not grow exponentially as the number of recurrences increases (Lemma 1). The main result in the paper is subsequently proved.

**Lemma 1.** *For input $(\boldsymbol{x}_0, \mathbf{u}_1)$ to a `MambaBlock`, small changes $(\boldsymbol{x}_0 + \varepsilon, \mathbf{u}_1 + \varepsilon)$ produce deviations which are exponentially non-increasing over discrete-time. That is, $\max |F_\theta^N(\boldsymbol{x}_0, \mathbf{u}_1) - F_\theta^N(\boldsymbol{x}_0 + \varepsilon, \mathbf{u}_1 + \varepsilon)| \in \mathcal{O}(\varepsilon \exp(N\zeta))$, for some scalar $\zeta \leqslant 0$.*

*Proof.* Firstly, we note that within the `MambaBlock`, $\mathbf{A}$ is stored in log-space followed by a negative exponentiation prior to use. Thus, $\mathbf{A} \in \mathbb{R}_{\leqslant 0}^{d \times d}$, where $\mathbb{R}_{\leqslant 0}$ is the set of non-positive real numbers.

Recall that for the maximum deviation, we have:

$$\max |F_\theta^N(\boldsymbol{x}_0, \mathbf{u}_1) - F_\theta^N(\boldsymbol{x}_0 + \varepsilon, \mathbf{u}_1 + \varepsilon)| \in \mathcal{O}(\varepsilon \exp(N\lambda_{\texttt{max}})).$$

where the maximal Lyapunov exponent $\lambda_{\max}$ is defined as:

$$\lambda_{\max} := \lim_{T \to \infty} \frac{1}{T} \log \left\| \prod_{t=0}^{T} \frac{\partial \boldsymbol{x}_t}{\partial \boldsymbol{x}_{t-1}} \right\|_2,$$

and $\|\|_2$ denotes the spectral norm for matrices.

Thus, to complete the proof, it suffices to show that $\lambda_{\max} \leqslant 0$. Recall that $\mathbf{A}$ and $\bar{\boldsymbol{\Delta}}_t$ are diagonal. From Equation 5, we thus have

$$\lambda_{\max} = \lim_{T \to \infty} \frac{1}{T} \log \left\| \prod_{t=0}^{T} \frac{\partial \boldsymbol{x}_t}{\partial \boldsymbol{x}_{t-1}} \right\|_2$$

$$= \lim_{T \to \infty} \frac{1}{T} \log \left\| \prod_{t=0}^{T} \exp\left(\bar{\boldsymbol{\Delta}}_t \mathbf{A}\right) \right\|_2$$

$$= \lim_{T \to \infty} \frac{1}{T} \log \left\| \exp \sum_{t=0}^{T} (\bar{\boldsymbol{\Delta}}_t \mathbf{A}) \right\|_2$$

Let $i$ be the dimension which corresponds to the output of the spectral norm, i.e., $i = \operatorname{argmax}_{j=1,\ldots,d}\{\exp \sum_{t=0}^{T}(\bar{\boldsymbol{\Delta}}_t[j,j]\mathbf{A}[j,j])\}$. We thus have

$$\lambda_{\max} = \lim_{T \to \infty} \frac{1}{T} \log \left\| \exp \sum_{t=0}^{T} (\bar{\boldsymbol{\Delta}}_t \mathbf{A}) \right\|_2$$

$$= \lim_{T \to \infty} \frac{1}{T} \log \exp \sum_{t=0}^{T} (\bar{\boldsymbol{\Delta}}_t[i,i]\mathbf{A}[i,i])$$

$$= \mathbf{A}[i,i] \lim_{T \to \infty} \frac{1}{T} \sum_{t=0}^{T} \bar{\boldsymbol{\Delta}}_t[i,i]$$

$\mathbf{A}[i,i]$ is non-positive and $\lim_{T \to \infty} \frac{1}{T} \sum_{t=0}^{T} \bar{\boldsymbol{\Delta}}_t[i,i] \geqslant 0$, since $\bar{\boldsymbol{\Delta}}_t[i,i] \in \mathbb{R}_{\geqslant 0} \ \forall t$. Thus, $\lambda_{\max} \leqslant 0$. $\square$

**Theorem 1.** *Let $(\boldsymbol{x}_{t-1}, \mathbf{u}_t)$ be the latent state and input at an arbitrary time $t \in [1, T]$ within a* `MambaBlock`. *Then small changes $(\boldsymbol{x}_{t-1} + \varepsilon, \mathbf{u}_t + \varepsilon)$ produce deviations which are exponentially decreasing over discrete-time, i.e., $\max |F_\theta^N(\boldsymbol{x}_0, \mathbf{u}_1) - F_\theta^N(\boldsymbol{x}_0 + \varepsilon, \mathbf{u}_1 + \varepsilon)| \in \mathcal{O}(\varepsilon \exp(N\zeta))$, for some scalar $\zeta \leqslant 0$.*

*Proof.* Let $\tau(t)$ be a function that maps time values such that $\tau(t) \in [1, T-t]$ and $\tau(t) = 1, \tau(t+1) = 2, \ldots, \tau(t+T) = T-t$. Then $\mathbf{B}_{\tau(t)}, \mathbf{C}_{\tau(t)}, \boldsymbol{\Delta}_{\tau(t)}$ define a new `MambaBlock` with inputs $\mathbf{u}_{\tau(t)}, \ldots, \mathbf{u}_{\tau(t+T)}$ and subsequent recurrent states $\boldsymbol{x}_{\tau(t)}, \ldots, \boldsymbol{x}_{\tau(t+T)}$. Applying Lemma 1 to this `MambaBlock` with $(\boldsymbol{x}_{\tau(t)-1}, \mathbf{u}_{\tau(t)})$ completes the proof. $\square$

## C  MAMBA STABLE OUTPUTS PROOF

**Theorem 2.** *Assume $(\boldsymbol{x}_{t-1} + \varepsilon, \mathbf{u}_t + \varepsilon)$ produce deviations which are exponentially non-increasing over discrete-time. Then small changes to the output $\mathbf{y}_t$ are also exponentially non-increasing over discrete time.*

*Proof.* Recall that $\boldsymbol{x}_T = F_\theta^T(\boldsymbol{x}_0, \mathbf{u}_1)$. Furthermore, recall from Equations 1 and 2, $\mathbf{y}_t = \mathbf{C}_t \boldsymbol{x}_t$, where $\mathbf{C}_t$ is diagonal.

Let

$$\mathbf{y}_T = G_\theta^T(\boldsymbol{x}_0, \mathbf{u}_1) = \mathbf{C}_T \boldsymbol{x}_T = \mathbf{C}_T F_\theta^T(\boldsymbol{x}_0, \mathbf{u}_1).$$

Consider $\varepsilon$-close inputs $(\boldsymbol{x}_{t-1}, \mathbf{u}_t)$ and $(\boldsymbol{x}_{t-1} + \varepsilon, \mathbf{u}_t + \varepsilon)$, and their respective outputs $\mathbf{y}_t$ and $\mathbf{y}'_t$. Assume $(\boldsymbol{x}_{t-1} + \varepsilon, \mathbf{u}_t + \varepsilon)$ produce deviations which are exponentially non-increasing over discrete-time. That is, $\max |F_\theta^N(\boldsymbol{x}_{t-1}, \mathbf{u}_t) - F_\theta^N(\boldsymbol{x}_{t-1} + \varepsilon, \mathbf{u}_t + \varepsilon)| \in \mathcal{O}(\varepsilon \exp(N\zeta))$, for some scalar $\zeta \leqslant 0$.

We thus have

$$
\begin{aligned}
\max |\mathbf{y}_t - \mathbf{y}'_t| &= \max |G_\theta^N(\boldsymbol{x}_{t-1}, \mathbf{u}_t) - G_\theta^N(\boldsymbol{x}_{t-1} + \varepsilon, \mathbf{u}_t + \varepsilon)| \\
&= \max |\mathbf{C}_N F_\theta^N(\boldsymbol{x}_{t-1}, \mathbf{u}_t) - \mathbf{C}_N F_\theta^N(\boldsymbol{x}_{t-1} + \varepsilon, \mathbf{u}_t + \varepsilon)| \\
&\propto \max |F_\theta^N(\boldsymbol{x}_{t-1}, \mathbf{u}_t) - F_\theta^N(\boldsymbol{x}_{t-1} + \varepsilon, \mathbf{u}_t + \varepsilon)|,
\end{aligned}
$$

where proportionality follows due to the diagonality of $\mathbf{C}_N$ and the vector-absolute value. Thus,

$$
\max |G_\theta^N(\boldsymbol{x}_{t-1}, \mathbf{u}_t) - G_\theta^N(\boldsymbol{x}_{t-1} + \varepsilon, \mathbf{u}_t + \varepsilon)| \in \mathcal{O}(\varepsilon \exp(N\zeta))
$$

$\square$

## D  PROOF OF WEIGHT-TYING USING LORA IN THE MAMBABLOCK

Due to the low-level nature of Mamba's prefix scan optimizations (discussed in Section 2), standard use of LoRA adapters is made difficult within Mamba's SSM-layer. E.g., while $B_t, C_t$ and $\Delta_t$ are conceptually `PyTorch` linear layers, their bundling in a contiguous memory block and careful manipulation makes appending a LoRA adapter on any of these individual matrices non-trivial (particularly, while respecting the highly specialized layout of each LoRA adapters targeted layer). However, we note that the overall design of the `MambaBlock`'s hardware optimizations may be leveraged to both efficiently learn the parameter-space for the majority of time-varying parameters (thus achieving PEFT) and regularize parameters during training (thus improving fine-tuning generalization).

**Theorem 3.** *Consider the weight matrix $\mathbf{W}$ of a `MambaBlock` from Equation 4. Targeting $\mathbf{W}$ for LoRA during fine-tuning ties adaptation weights across $\mathbf{B}_t, \mathbf{C}_t$ and $\boldsymbol{\Delta}_t$.*

*Proof.* Let $r$ be the specified LoRA dimension. Targeting this matrix for LoRA results in the adapter

$$
\begin{aligned}
\tilde{\mathbf{W}} &= \mathbf{W} + \mathbf{W}' \\
&= \mathbf{W} + \mathbf{U}\mathbf{V},
\end{aligned}
$$

where $\mathbf{U} \in \mathbb{R}^{n \times r}, \mathbf{V} \in \mathbb{R}^{r \times 3d}$, and $\mathbf{W}$ is frozen during fine-tuning. Thus, for index $[i, j]$,

$$
\mathbf{W}'[i, j] = \sum_{k=0}^{r-1} \mathbf{U}[i, k]\mathbf{V}[k, j].
$$

Recall the form of $\mathbf{W}$:

$$
\mathbf{W}[t-1, :d] = \mathtt{diag}(\boldsymbol{\Delta}_t), \mathbf{W}[t-1, d:2d] = \mathtt{diag}(\mathbf{B}_t), \mathbf{W}[t-1, 2d:3d] = \mathtt{diag}(\mathbf{C}_t),
$$

where $\mathbf{W}[0, :d] = \mathtt{diag}(\boldsymbol{\Delta}_1), \mathbf{W}[n-1, d:2d] = \mathtt{diag}(\mathbf{B}_T)$, and so on. For index $[t-1, j]$, we thus have

$$
\begin{aligned}
\tilde{\mathbf{W}}[t-1, j] &= \mathbf{W}[t-1, j] + \mathbf{W}'[t-1, j] \\
&= \mathbf{W}[t-1, j] + \sum_{k=0}^{r-1} \mathbf{U}[t-1, k]\mathbf{V}[k, j].
\end{aligned}
$$

Thus, the weights $\mathbf{U}[t-1, :]$ are tied for any parameter $\tilde{\mathbf{W}}[t-1, j], j \in \{1, \ldots, 3d\}$, which are used to adapt parameters $\boldsymbol{\Delta}_1, \mathbf{B}_t$, and $\mathbf{C}_t$.

$\square$

# E    EXPERIMENTAL DETAILS

All model checkpoints were evaluated on all benchmarks and few-shot settings using the LM evaluation harness from Eleuther AI (Gao et al., 2023), version `0.4.2`. Pythia and Mamba `Huggingface` checkpoints were used for all inference and fine-tuning experiments, e.g., `EleutherAI/pythia-160m` and `state-spaces/mamba-130m-hf` for the smallest respective models. All fine-tuning experiments were run using package versions `Transformers 4.40.0.dev0`, `Accelerate 0.28.0`, `TRL 0.8.1`, `PyTorch 2.2.1+cu121`, and `PEFT 0.10.0`. All Mamba-2 models were run using `mamba-ssm v2.2.2` using `Huggingface` checkpoints, e.g., `state-spaces/mamba-130m` for the smallest model.

For MPFT, `Flash Attention 2.0` (Dao et al., 2022) via `flash_attn 2.5.7` was used for Pythia models. For `FP16` and `BF16` inference results, Flash Attention 2.0 was used for both Pythia and OLMo models. For OLMo results, the 336B-token checkpoint was used by specifying `revision=step80000-tokens336B`.

All Alpaca and OpenHermes fine-tuning experiments used the following training recipe (adapted from (Tunstall et al., 2023)): `AdamW_torch` optimizer, `cosine annealing` schedule, no gradient accumulation, maximum norm of 1.0 for gradient clipping, and no warmup steps. Training epochs used for all Alpaca and OpenHermes experiments were three and one, respectively. For both Pythia and Mamba models, the learning rate and LoRA dimension $r$ were scaled to improve performance of smaller models (per-model values listed in Table 3).

For `SLL LoRA`, targeted Mamba layers were `{x_proj, embeddings, in_proj, out_proj}`; `x_proj` is the large `MambaBlock` memory buffer which, when targeted by LoRA, regularizes the majority of SSM parameters during fine-tuning through weight tying (Theorem 3). Pythia targeted `SLL LoRA` layers were `{dense, embed_in, query_key_value, dense_h_to_4h, dense_4h_to_h}`, chosen to balance performance across model sizes.

All experiments were run using a single-GPU Nvidia A10G (24 GB total memory). For Pythia, Mamba, and Mamba-2 `ALL LoRA` experiments in Figure 3, all models followed the same training and PEFT recipes, save for Mamba-2 `2.7B` which required a LoRA $r$ dimension of 64 to fit in A10G memory.

Table 3: Learning rate and LoRA dimension $r$ values

| Mamba size | Mamba-2 size | Pythia size | learning rate | Mamba/Pythia LoRA $r$ | Mamba-2 LoRA $r$ |
|---|---|---|---|---|---|
| 130M | 130M | 160M | 1.0e-5 | 8 | 8 |
| 370M | 370M | 410M | 5.0e-5 | 16 | 16 |
| 790M | 780M | 1B | 1.0e-6 | 32 | 32 |
| 1.4B | 1.3B | 1.4B | 5.0e-6 | 64 | 64 |
| 2.8B | 2.7B | 2.8B | 5.0e-7 | 128 | 64 |

The Alpaca dataset is freely available for download at `https://huggingface.co/datasets/tatsu-lab/alpaca` under open-source license `CC-by-NC 4.0`. The OpenHermes dataset is freely available for download at `https://huggingface.co/datasets/teknium/OpenHermes-2.5` under open-source license `MIT, Apache 2.0, CC`.

# F    MAMBA-2 MIXED-PRECISION INFERENCE PRETRAINED LLM PERFORMANCE TABLES

Table 4: Mean full-precision (`FP32`) divergence in MMLU performance for mixed-precision inference. Divergence is averaged over {0, 1, 3, 5}-shot performance. Pretrained checkpoints are used for Mamba (`M`), Mamba-2 (`M2`), Pythia (`P`), OLMo (Groeneveld et al., 2024) (`O`), and Phi-1.5 (Li et al., 2023) (`Phi`) models.

| Model | M | M2 | P | M | M2 | P | M | M2 | P | O | M | M2 | P | Phi | M | M2 | P |
|---|---|---|---|---|---|---|---|---|---|---|---|---|---|---|---|---|---|
| Size | 130m | 130m | 160m | 370m | 370m | 410m | 790m | 780m | 1b | | 1.4b | 1.3b | 1.4b | 1.5b | 2.8b | 2.7b | 2.8b |
| FP16 $\mu$ | 0.03 | 0.07 | 0.35 | 0.05 | 0.05 | 0.06 | 0.21 | 0.12 | 0.05 | 0.04 | 0.04 | 0.04 | 0.07 | 0.03 | 0.15 | 0.26 | 0.12 |
| BF16 $\mu$ | 0.05 | 0.67 | 1.45 | 0.20 | 0.52 | 0.20 | 0.66 | 0.29 | 0.16 | 0.13 | 0.31 | 0.40 | 0.13 | 1.05 | 1.17 | 1.02 | 0.11 |

# G    ADAPTED PEFT-LAYER EFFECTS ON MAMBA-2 ZERO-SHOT PERFORMANCE

Table 5: **Pretrained model performance**. Model checkpoints were evaluated on all benchmarks and few-shot settings using the LM evaluation harness from Eleuther AI (Gao et al., 2023). LAMBADA zero-shot is more effective for the model sizes considered (further discussed in (Xie et al., 2021; Brown et al., 2020)) and thus excluded from few-shot performance averages. Highlighted in bold is the top-performing few-shot learner per benchmark and model grouping.

| Model | N-shot | LAMBADA ppl ↓ | LAMBADA acc ↑ | HellaSwag acc ↑ | PIQA acc ↑ | Arc-E acc ↑ | Arc-C acc ↑ | WinoGrande acc ↑ | 0-shot incr. Mean % ↑ |
|---|---|---|---|---|---|---|---|---|---|
| Mamba 130M | 0 | **16.0** | **44.3** | 35.2 | 64.7 | 48.0 | 24.3 | 52.6 | – |
| | 1 | 19.3 | 38.2 | 35.1 | 64.6 | 47.0 | 23.5 | 50.8 | -1.9 |
| | 3 | 23.1 | 35.2 | 35.0 | 65.1 | 49.1 | 24.0 | 51.0 | -0.4 |
| | 5 | 24.4 | 36.2 | 34.9 | 64.9 | 49.1 | 23.7 | 50.0 | -1.2 |
| Mamba-2 130M | 0 | 16.8 | 43.9 | 35.3 | 64.9 | 47.4 | 24.2 | 52.6 | – |
| | 1 | 20.6 | 37.9 | 34.9 | 64.1 | 46.9 | 23.1 | 51.3 | -2.0 |
| | 3 | 24.3 | 35.1 | 34.9 | 64.4 | **49.0** | 24.7 | **52.9** | **0.9** |
| | 5 | 26.5 | 34.9 | 34.6 | 64.4 | 48.6 | **24.8** | 51.7 | 0.2 |
| Pythia 160M | 0 | 38.2 | 32.7 | 30.2 | 61.8 | 43.4 | 23.8 | 51.0 | – |
| | 1 | 47.2 | 28.2 | 30.6 | 62.2 | 43.4 | 23.7 | 49.3 | -0.4 |
| | 3 | 63.7 | 24.7 | 30.5 | 61.9 | 44.8 | 22.9 | 51.3 | 0.1 |
| | 5 | 66.3 | 25.3 | 30.4 | 62.6 | 43.4 | 23.1 | 50.8 | -0.2 |
| Mamba 370M | 0 | 8.1 | 55.6 | 46.5 | 69.5 | 54.9 | 27.8 | 55.3 | – |
| | 1 | 9.7 | 49.8 | 45.9 | 69.3 | 57.4 | 26.5 | 54.7 | -0.5 |
| | 3 | 10.9 | 48.4 | 46.2 | 69.5 | 58.8 | 28.4 | 53.8 | 1.2 |
| | 5 | 11.4 | 48.6 | 46.2 | 69.4 | 58.3 | 28.0 | **55.9** | 1.5 |
| Mamba-2 370M | 0 | **8.0** | **55.9** | **46.9** | **70.5** | 54.8 | 26.7 | 55.4 | – |
| | 1 | 9.8 | 50.3 | 46.4 | **70.5** | 56.5 | 26.8 | 54.2 | 0.0 |
| | 3 | 11.3 | 48.5 | 46.6 | 70.2 | **59.0** | 26.9 | 54.3 | 1.0 |
| | 5 | 12.5 | 46.6 | 46.7 | 70.3 | 58.5 | 28.2 | 53.3 | 1.5 |
| Pythia 410M | 0 | 10.8 | 51.5 | 40.6 | 66.9 | 52.0 | 24.1 | 53.4 | – |
| | 1 | 12.3 | 47.1 | 40.5 | 68.0 | 53.8 | 25.6 | 52.4 | 1.8 |
| | 3 | 14.4 | 43.2 | 40.9 | 67.9 | 55.1 | 26.9 | 54.0 | **4.2** |
| | 5 | 14.6 | 44.1 | 40.8 | 68.1 | 54.6 | 26.6 | 53.4 | 3.5 |
| Mamba 790M | 0 | 6.0 | 61.4 | **55.1** | 72.3 | 61.2 | 29.5 | 55.9 | – |
| | 1 | 7.1 | 55.9 | 54.5 | 72.4 | 63.0 | 30.2 | 56.9 | 1.3 |
| | 3 | 8.1 | 54.5 | 54.2 | 72.3 | 63.5 | 31.4 | 57.1 | 2.2 |
| | 5 | 8.8 | 52.9 | 54.6 | **72.6** | 64.4 | 31.9 | 57.2 | **3.1** |
| Mamba-2 780M | 0 | **5.9** | **61.7** | 54.9 | 72.0 | 61.0 | 28.5 | **60.2** | – |
| | 1 | 7.1 | 55.5 | 54.7 | 72.4 | 62.3 | 32.1 | 57.1 | 1.9 |
| | 3 | 8.6 | 53.3 | 54.7 | 72.5 | 62.8 | **32.3** | 57.8 | 2.5 |
| | 5 | 9.9 | 51.4 | 55.2 | 72.1 | 62.8 | 32.2 | 56.8 | 2.2 |
| Pythia 1B | 0 | 7.9 | 56.3 | 47.2 | 70.7 | 57.0 | 27.0 | 53.4 | – |
| | 1 | 8.0 | 51.8 | 47.3 | 70.7 | 57.1 | 28.2 | 53.4 | 1.0 |
| | 3 | 10.5 | 48.2 | 47.5 | 71.2 | 59.2 | 28.0 | 54.3 | 2.2 |
| | 5 | 10.9 | 48.4 | 47.3 | 71.4 | 58.7 | 28.4 | 53.1 | 1.9 |
| Mamba 1.4B | 0 | **5.0** | 64.9 | 59.2 | 74.1 | 65.5 | 32.9 | 61.3 | – |
| | 1 | 5.8 | 60.6 | 58.2 | **74.7** | 64.5 | 33.0 | 60.9 | -0.5 |
| | 3 | 6.6 | 58.9 | 58.9 | 73.6 | 66.1 | 34.5 | 60.9 | 0.7 |
| | 5 | 7.0 | 58.3 | 59.0 | 74.1 | 66.4 | 35.5 | 60.4 | 1.5 |
| Mamba-2 1.3B | 0 | **5.0** | **65.6** | 60.0 | 73.2 | 64.2 | 33.1 | 61.1 | – |
| | 1 | 6.0 | 60.1 | 59.4 | 73.1 | 65.6 | 35.3 | 59.4 | 1.0 |
| | 3 | 6.7 | 58.6 | 60.1 | 73.4 | **66.5** | 35.4 | **61.9** | 2.5 |
| | 5 | 7.0 | 58.6 | **60.2** | 73.7 | 66.5 | **35.9** | 61.4 | 2.7 |
| Pythia 1.4B | 0 | 6.1 | 61.7 | 52.1 | 70.9 | 60.5 | 28.5 | 57.4 | – |
| | 1 | 7.0 | 56.3 | 52.1 | 71.4 | 62.0 | 29.5 | 57.5 | 1.4 |
| | 3 | 7.9 | 54.4 | 52.6 | 70.9 | 63.9 | 31.1 | 56.8 | 2.9 |
| | 5 | 8.0 | 54.4 | 52.8 | 71.0 | 63.2 | 31.3 | 57.8 | **3.3** |
| Mamba 2.8B | 0 | 4.2 | 69.1 | 66.1 | 75.2 | 69.6 | 36.4 | 63.3 | – |
| | 1 | 5.0 | 63.7 | 65.6 | 75.6 | 69.9 | 37.1 | 63.9 | 0.6 |
| | 3 | 5.5 | 62.8 | 65.5 | 75.3 | 70.8 | 38.1 | 65.1 | 1.7 |
| | 5 | 5.7 | 62.5 | 66.1 | 76.1 | 70.9 | 38.1 | 64.6 | 2.0 |
| Mamba-2 2.7B | 0 | **4.1** | **69.6** | 66.6 | **76.4** | 69.5 | 36.3 | 63.9 | – |
| | 1 | 4.8 | 65.1 | 65.9 | 75.1 | 70.0 | 38.6 | 65.1 | 1.3 |
| | 3 | 5.3 | 63.9 | 66.8 | 75.2 | **71.9** | 41.0 | 64.1 | 3.1 |
| | 5 | 5.7 | 62.3 | **67.1** | 75.3 | 70.7 | **41.2** | **65.9** | **3.6** |
| Pythia 2.8B | 0 | 5.0 | 64.7 | 59.3 | 73.9 | 64.2 | 32.9 | 59.8 | – |
| | 1 | 5.7 | 60.9 | 59.4 | 73.8 | 66.8 | 34.8 | 59.0 | 1.7 |
| | 3 | 6.2 | 59.1 | 59.9 | 74.7 | 67.4 | 34.9 | 60.8 | 2.9 |
| | 5 | 6.5 | 59.1 | 60.2 | 74.5 | 67.1 | 35.0 | 61.3 | 3.1 |

Table 6: **Instruction tuned model performance**. Model checkpoints were evaluated on all benchmarks and few-shot settings using the LM evaluation harness from Eleuther AI (Gao et al., 2023). LAMBADA zero-shot is more effective for the model sizes considered (further discussed in (Xie et al., 2021; Brown et al., 2020)) and thus excluded from few-shot performance averages. Highlighted in bold is the top-performing few-shot learner per benchmark and model grouping.

| Model | N-shot | LAMBADA ppl ↓ | LAMBADA acc ↑ | HellaSwag acc ↑ | PIQA acc ↑ | Arc-E acc ↑ | Arc-C acc ↑ | WinoGrande acc ↑ | 0-shot incr. Mean % ↑ |
|---|---|---|---|---|---|---|---|---|---|
| | 0 | **12.9** | **46.5** | **35.1** | 64.2 | 48.7 | **25.5** | 51.7 | – |
| Mamba | 1 | 17.8 | 38.1 | 35.0 | 64.2 | 48.6 | 24.9 | **52.2** | -0.4 |
| 130M | 3 | 22.3 | 35.3 | 34.8 | 64.2 | **50.2** | 24.5 | 50.6 | -0.8 |
| | 5 | 23.6 | 35.9 | 34.7 | 64.7 | 49.8 | 24.6 | 50.2 | -0.9 |
| | 0 | 15.2 | 44.5 | **35.1** | 64.5 | 47.2 | 24.7 | **52.2** | – |
| Mamba-2 | 1 | 21.9 | 36.1 | 34.5 | 64.3 | 46.8 | 24.0 | 50.8 | -1.7 |
| 130M | 3 | 26.9 | 33.3 | 34.7 | **65.1** | 48.5 | 25.2 | 51.5 | **0.6** |
| | 5 | 29.0 | 33.8 | 34.5 | 64.8 | 48.7 | 25.1 | 51.3 | 0.4 |
| | 0 | 30.2 | 36.1 | 30.0 | 62.2 | 44.7 | 23.6 | 50.3 | – |
| Pythia | 1 | 44.5 | 29.1 | 30.4 | 62.0 | 44.0 | 23.6 | 50.5 | -0.0 |
| 160M | 3 | 66.7 | 25.5 | 30.3 | 62.8 | 45.2 | 22.8 | 49.8 | -0.3 |
| | 5 | 70.4 | 25.3 | 30.5 | 62.9 | 44.1 | 23.4 | 50.8 | 0.3 |
| | 0 | 7.2 | **56.0** | 46.3 | 69.2 | 55.3 | 27.7 | **56.0** | – |
| Mamba | 1 | 9.3 | 49.9 | 45.7 | 68.7 | 57.1 | 28.3 | 55.4 | 0.5 |
| 370M | 3 | 10.4 | 49.4 | 45.7 | 68.9 | 58.7 | **29.7** | 54.1 | 1.6 |
| | 5 | 11.0 | 48.3 | 45.7 | 70.1 | 59.3 | 29.1 | 54.5 | 1.9 |
| | 0 | **7.6** | 54.7 | **46.8** | 69.3 | 52.2 | 27.0 | **56.0** | – |
| Mamba-2 | 1 | 9.9 | 48.3 | 46.0 | 69.6 | 55.7 | 28.8 | 55.2 | 2.1 |
| 370M | 3 | 11.8 | 46.3 | 46.3 | 70.1 | 59.0 | 29.1 | 54.5 | 3.6 |
| | 5 | 12.6 | 45.5 | 46.3 | **70.8** | **59.6** | 29.5 | 53.0 | **3.8** |
| | 0 | 13.3 | 46.4 | 40.9 | 67.4 | 52.7 | 25.4 | 53.4 | – |
| Pythia | 1 | 17.2 | 40.4 | 40.5 | 68.4 | 53.6 | 25.7 | 53.0 | 0.5 |
| 410M | 3 | 21.1 | 37.4 | 40.9 | 67.7 | 55.7 | 27.1 | 52.6 | 2.3 |
| | 5 | 21.5 | 38.2 | 40.7 | 67.8 | 55.7 | 27.3 | 53.8 | 2.8 |
| | 0 | 5.2 | 62.8 | 55.6 | 72.8 | 62.4 | 30.6 | 56.2 | – |
| Mamba | 1 | 6.3 | 56.6 | 54.9 | 72.7 | 64.6 | 31.7 | 56.3 | 1.2 |
| 790M | 3 | 7.0 | 55.6 | 54.7 | 72.4 | **65.3** | 33.2 | 57.5 | 2.7 |
| | 5 | 7.5 | 54.6 | 54.9 | 72.9 | 65.6 | 33.8 | 57.2 | 3.2 |
| | 0 | **4.9** | **63.4** | **55.8** | 71.7 | 61.1 | 30.6 | **59.2** | – |
| Mamba-2 | 1 | 6.6 | 55.2 | 54.4 | 72.7 | 64.2 | 34.0 | 57.6 | 2.5 |
| 780M | 3 | 7.8 | 52.7 | 54.9 | **73.5** | 65.0 | **34.6** | 57.8 | **3.6** |
| | 5 | 8.6 | 52.8 | 54.8 | 73.4 | 64.6 | 34.0 | 58.0 | 3.1 |
| | 0 | 7.7 | 56.6 | 47.3 | 70.8 | 57.1 | 26.7 | 53.4 | – |
| Pythia | 1 | 8.8 | 52.0 | 47.4 | 70.7 | 57.5 | 28.8 | 53.6 | 1.8 |
| 1B | 3 | 10.2 | 48.7 | 47.5 | 71.4 | 59.0 | 28.5 | 54.4 | 2.6 |
| | 5 | 10.6 | 48.8 | 47.4 | 71.5 | 58.9 | 28.4 | 53.0 | 2.0 |
| | 0 | **4.6** | **64.8** | 59.3 | 74.3 | 65.2 | 35.1 | 62.3 | – |
| Mamba | 1 | 5.4 | 60.3 | 58.2 | 74.3 | 66.7 | 35.7 | **62.8** | 0.6 |
| 1.4B | 3 | 6.1 | 59.3 | 58.4 | 74.1 | 67.4 | 36.6 | 61.8 | 1.0 |
| | 5 | 6.3 | 58.8 | 58.8 | **74.5** | 68.3 | **37.0** | 59.9 | 1.1 |
| | 0 | 4.9 | 63.0 | **60.1** | 73.8 | 64.0 | 34.8 | 61.3 | – |
| Mamba-2 | 1 | 6.1 | 58.2 | 59.2 | 74.2 | 67.0 | 35.0 | 60.1 | 0.5 |
| 1.3B | 3 | 7.0 | 56.6 | 59.4 | 73.7 | 67.8 | 36.6 | 59.9 | 1.5 |
| | 5 | 7.2 | 56.5 | 59.9 | 73.5 | **68.5** | 36.7 | 60.7 | 2.2 |
| | 0 | 5.2 | 63.6 | 52.9 | 71.1 | 61.2 | 30.3 | 58.2 | – |
| Pythia | 1 | 6.2 | 57.4 | 52.7 | 71.7 | 62.2 | 30.6 | 56.9 | 0.2 |
| 1.4B | 3 | 7.0 | 56.1 | 53.1 | 71.1 | 64.5 | 32.8 | 56.8 | 2.3 |
| | 5 | 7.1 | 55.5 | 53.3 | 71.2 | 63.8 | 33.5 | 57.5 | **2.9** |
| | 0 | 4.0 | 67.7 | 66.4 | 75.6 | 68.4 | 36.6 | 64.2 | – |
| Mamba | 1 | 4.8 | 63.3 | 65.9 | 76.2 | 70.9 | 39.4 | 64.6 | 2.4 |
| 2.8B | 3 | 5.3 | 62.1 | 65.7 | 75.8 | 71.3 | 39.1 | 65.4 | 2.4 |
| | 5 | 5.4 | 61.9 | 66.2 | 77.2 | 71.4 | 40.4 | 66.1 | 3.9 |
| | 0 | **3.8** | **68.4** | **67.5** | 76.0 | 69.5 | 38.3 | 65.3 | – |
| Mamba-2 | 1 | 4.5 | 63.8 | 66.7 | 76.0 | 71.8 | 41.5 | **67.1** | 2.6 |
| 2.7B | 3 | 5.0 | 62.3 | 67.3 | 76.2 | **73.3** | 44.4 | 66.0 | 4.5 |
| | 5 | 5.3 | 61.8 | 67.4 | **76.4** | 72.4 | **44.5** | 65.0 | **4.1** |
| | 0 | 5.0 | 64.7 | 59.3 | 74.0 | 64.7 | 33.3 | 59.2 | – |
| Pythia | 1 | 5.6 | 60.8 | 59.5 | 74.0 | 66.7 | 34.9 | 59.3 | 1.7 |
| 2.8B | 3 | 6.1 | 59.2 | 59.9 | 75.0 | 67.5 | 34.9 | 60.9 | 2.9 |
| | 5 | 6.5 | 59.0 | 60.4 | 74.5 | 67.0 | 35.1 | 61.2 | 3.0 |

Table 7: Zero-shot performance for instruction tuned Mamba-2 models. ✓ denotes only **W** was targeted for LoRA adaptation within the `MambaBlock`, and ✗ thus denotes both linear layers within each `MambaBlock` were adapted. The top-performance for each task per model is highlighted in bold.

| Model | Only **W** targeted? | LAMBADA ppl ↓ | LAMBADA acc ↑ | HellaSwag acc ↑ | PIQA acc ↑ | Arc-E acc ↑ | Arc-C acc ↑ | WinoGrande acc ↑ |
|-------|-----|---------|---------|-----------|------|-------|-------|-----------|
| Mamba-2 | ✓ | **4.54** | **65.73** | **60.88** | 73.67 | **66.16** | 34.56 | **61.80** |
| 1.3B | ✗ | 4.94 | 62.95 | 60.06 | **73.78** | 64.02 | **34.81** | 61.33 |
| Mamba-2 | ✓ | 4.05 | **69.63** | 66.73 | **76.50** | **69.95** | 36.77 | 64.56 |
| 2.7B | ✗ | **3.77** | 68.39 | **67.51** | 75.95 | 69.53 | **38.31** | **65.35** |

## H  INSTRUCTION TUNING ROBUSTNESS

We show that Mamba is robust to the choice of PEFT hyperparemters. We conduct an extensive hyperparameter search across the learning rate, LoRA dimension, and number of warmup steps. From the Cartesian-product of these three parameters, 150 hyperparameter configurations were sampled and used to fine-tune `Mamba 370M` over the Openhermes dataset. For comparison, `Pythia 410M` is similarly fine-tuned using the same set of 150 hyperparameter configurations.

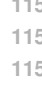
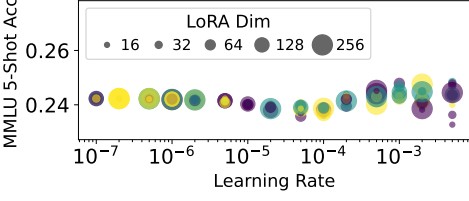
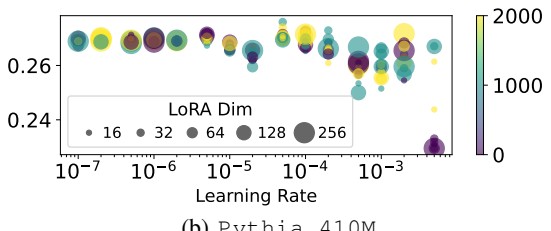

(a) `Mamba 370M`                    (b) `Pythia 410M`

Figure 5: Fine-tuning hyperparameter search for OpenHermes. Each point is a different hyperparameter configuration. `SLL LoRA` was used for both models. The $x$-axis is the learning rate, the $y$-axis is resulting MMLU 5-shot performance, bubble size is the LoRA dimension, and the color is the number of warmup steps $\in \{0, 1k, 2k\}$.

The MMLU 5-shot performance for each of the 150 Mamba and Pythia fine-tuned models is displayed in Figure 5. `Pythia 410M` is capable of higher performance than `Mamba 370M`, where the average accuracy for the former and the latter are 26.5% and 24.8%, respectively. However, `Mamba 370M` is much more robust to the choice of hyperparameters, with a difference of 1.5% between the minimum (23.3%) and maximum (24.8%). In contrast, `Pythia 410M` fine-tuned models display a large performance difference of 4.7% between the minimum (22.9%) and maximum (27.6%).

## I  EXPANDED DIVERGENCE RESULTS: ALPACA AND LIMA FINE-TUNING, MMLU AND WINOGRANDE BENCHMARKS, MEAN AND STANDARD DEVIAATION DIVERGENCES

We extend the non-divergent Mamba fine-tuning results from Section 4. Recall that the following MPFT and PEFT configurations are considered to fine-tune each considered LLM:

1. Full fine-tuning in `FP32`
2. Full fine-tuning in `FP16`
3. Full fine-tuning in `BF16`
4. `ALL LoRA`fine-tuning in `FP32`
5. `ALL LoRA`fine-tuning in `FP16`
6. `ALL LoRA`fine-tuning in `BF16`

7. SLL LoRA fine-tuning in FP32

8. SLL LoRA fine-tuning in FP16

9. SLL LoRA fine-tuning in BF16

In addition to the Alpaca dataset (Taori et al., 2023), we also fine-tune all models using the LIMA dataset (Zhou et al., 2024). Models are trained using LIMA for 5 epochs, while all other settings follow the fine-tuning recipe used for Alpaca (described in Appendix E).

For natural language benchmarks, in addition to MMLU, we evaluate each fine-tuned model using Winogrande (Sakaguchi et al., 2021). Recall that, for each benchmark, divergence between a mixed-precision fine-tuned model is measured between its full-precision counterpart and averaged over {0, 1, 3, 5}-shot performance. In addition to the average divergence, we also include the standard deviation of divergence. Thus, **in total, 72 new LLMs were fine-tuned, while 360 new MMLU and 720 new Winogrande evaluations were conducted, respectively**.

Figure I displays results for Alpaca and MMLU, Figure I displays results for Alpaca and Winogrande, Figure I displays results for LIMA and MMLU, Figure I displays results for LIMA and Winogrande. Summary statistics for all experiments are presented in Table I. While OpenELM exhibits large deviation spikes for both Alpaca benchmark evaluations–and Pythia exhibits large deviation spikes for all four evaluations–**Mamba does not exhibit a single large deviation spike on any benchmark for all considered model sizes and MPFT/PEFT configurations** (i.e., 18 total configurations excluding the full-precision baselines). Furthermore, Mamba models are significantly more stable for MPFT/PEFT compared to Transformer-based LLMs. E.g., **for MMLU evaluations, Alpaca fine-tuning with Mamba models is an average 2.6 times smaller in mean divergence than both Pythia and OpenELM models, while LIMA fine-tuning with Mamba models is an average 7 and 3.25 times smaller in mean divergence than Pythia and OpenELM models, respectively**.

Table 8: Summary of divergence results for Alpaca and LIMA fine-tuning datasets, MMLU and Winogrande benchmarks, and Mamba, OpenELM, and Pythia models. For each deviation summary statistic per fine-tuning dataset and benchmark, the lowest deviation is highlighted in bold.

| (Fine-tuning dataset), Benchmark | Architecture | Large deviation spikes ↓ | Avg mean divergence ↓ | Std mean divergence ↓ |
|---|---|---|---|---|
| (Alpaca, MMLU) | Pythia | 1 | 0.37 | 0.41 |
| | OpenELM | 1 | 0.37 | 0.32 |
| | Mamba | **0** | **0.14** | **0.08** |
| (Alpaca, Winogrande) | Pythia | 4 | 0.72 | 0.58 |
| | OpenELM | 3 | 0.59 | 0.37 |
| | Mamba | **0** | **0.25** | **0.09** |
| (LIMA, MMLU) | Pythia | 1 | 0.28 | 0.34 |
| | OpenELM | **0** | 0.13 | 0.15 |
| | Mamba | **0** | **0.04** | **0.03** |
| (LIMA, Winogrande) | Pythia | 3 | 0.45 | 0.45 |
| | OpenELM | **0** | 0.36 | 0.18 |
| | Mamba | **0** | **0.11** | **0.12** |

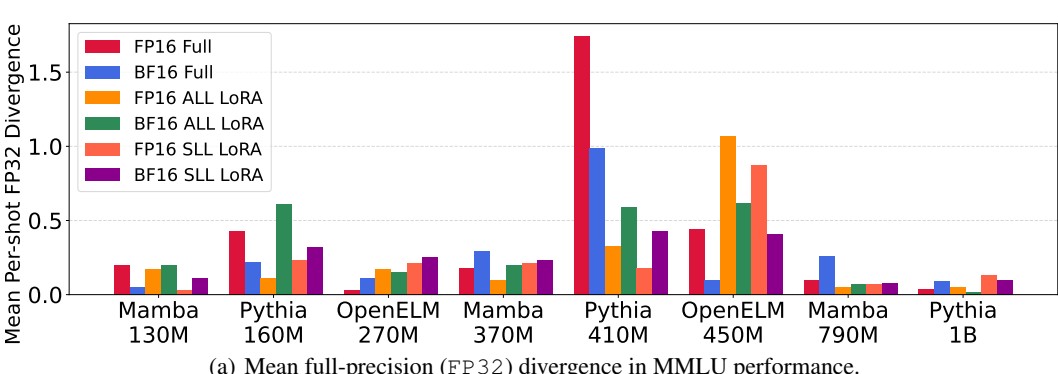

(a) Mean full-precision (`FP32`) divergence in MMLU performance.

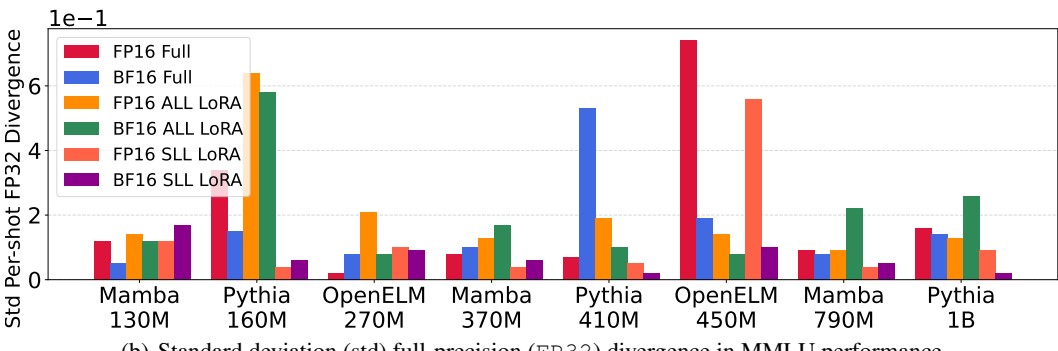

(b) Standard deviation (std) full-precision (`FP32`) divergence in MMLU performance.

Figure 6: **Alpaca fine-tuning, MMLU evaluation.** Mamba, Pythia, and OpenELM models are fine-tuned over the `Alpaca` dataset using different combinations of MPFT and PEFT. Full fine-tuning (i.e., no PEFT adapters) is denoted as `Full`.

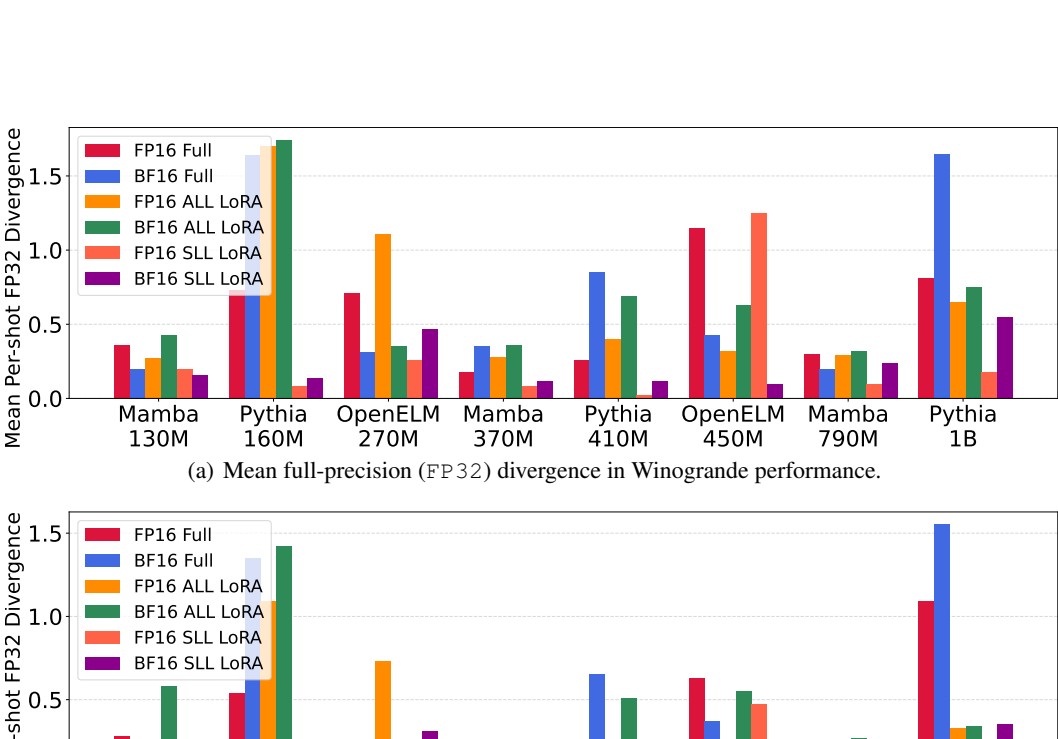

(a) Mean full-precision (`FP32`) divergence in Winogrande performance.

(b) Standard deviation (std) full-precision (`FP32`) divergence in Winogrande performance.

Figure 7: **Alpaca fine-tuning, Winogrande evaluation.** Mamba, Pythia, and OpenELM models are fine-tuned over the `Alpaca` dataset using different combinations of MPFT and PEFT. Full fine-tuning (i.e., no PEFT adapters) is denoted as `Full`.

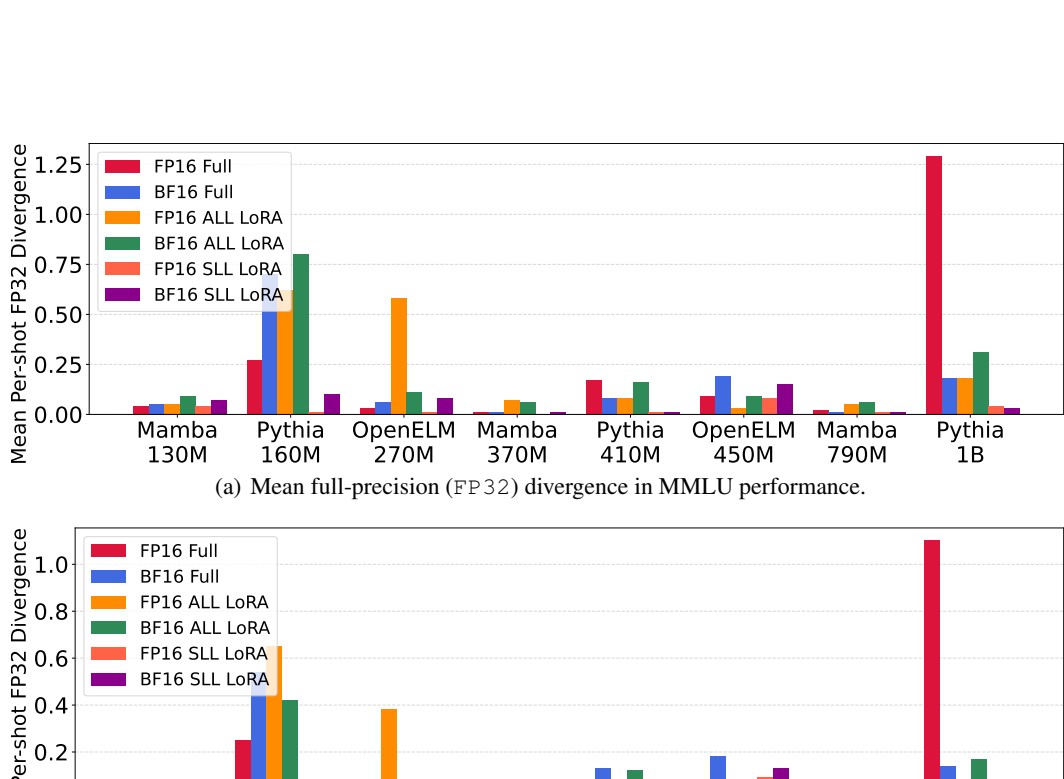

(a) Mean full-precision (FP32) divergence in MMLU performance.

(b) Standard deviation (std) full-precision (FP32) divergence in MMLU performance.

Figure 8: **LIMA fine-tuning, MMLU evaluation.** Mamba, Pythia, and OpenELM models are fine-tuned over the LIMA dataset using different combinations of MPFT and PEFT. Full fine-tuning (i.e., no PEFT adapters) is denoted as Full.

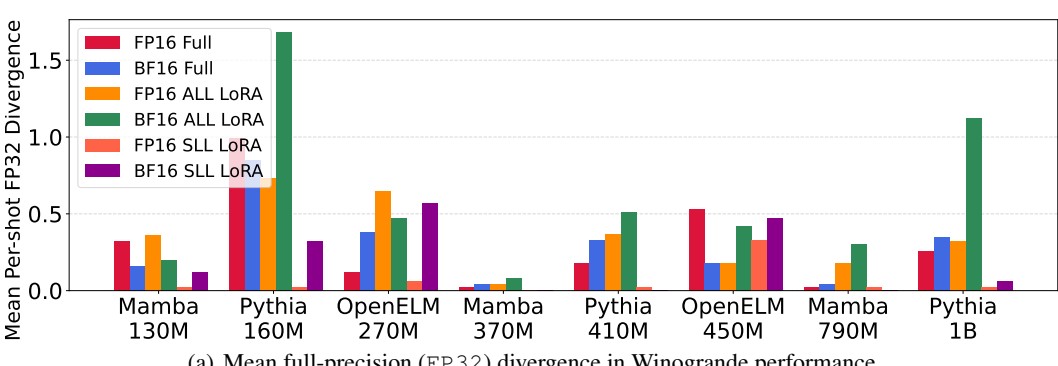

(a) Mean full-precision (`FP32`) divergence in Winogrande performance.

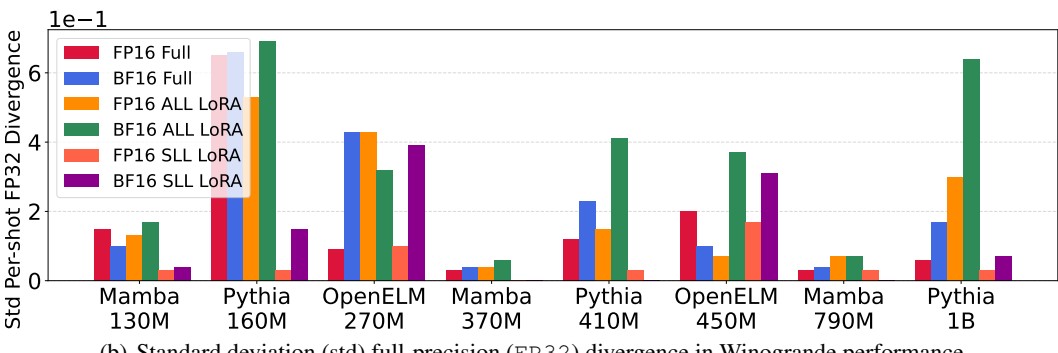

(b) Standard deviation (std) full-precision (`FP32`) divergence in Winogrande performance.

Figure 9: **LIMA fine-tuning, Winogrande evaluation.** Mamba, Pythia, and OpenELM models are fine-tuned over the `LIMA` dataset using different combinations of MPFT and PEFT. Full fine-tuning (i.e., no PEFT adapters) is denoted as `Full`.

