# OpenReview forum: "MAMBA STATE-SPACE MODELS ARE LYAPUNOV-STABLE LEARNERS"
_ICLR.cc/2025/Conference — Submitted to ICLR 2025_

### Official Review · Reviewer_jrwp · 2024-11-03

**Soundness:** 2
**Presentation:** 2
**Contribution:** 2
**Rating:** 3
**Confidence:** 5

**Summary:**

This paper studies Mamba, a state-space model (SSM) that has previously been introduced as an efficient alternative to Transformers. The authors analyze Mamba's stability in mixed-precision and parameter-efficient fine-tuning by applying Lyapunov exponents, theoretically and empirically confirming its resilience against input perturbations. Experiments show that Mamba maintains stability better than comparable Transformers under mixed precision and achieves notable memory and computation efficiency. Additionally, instruction tuning enhances Mamba's in-context learning, suggesting that SSMs can approach or surpass Transformer models in certain few-shot tasks.

**Strengths:**

This work provides promising evidence that fine-tuning Mamba on instruction-tuning data can improve its in-context learning abilities.

**Weaknesses:**

- Line 51:  For some training framework it does not keep master weight in fp32 which problematic for Mamba. But Mamba was still trained with mixed precision using AMP.

- Theory 1 seems shallow: Isn't Lemma 1 trivial to derive? \( dx_{t+1}/dx_t \) directly represents the decay rate, which must be \(\leq 1\) to prevent exponential growth in cumulative products. This insight is obvious for any recurrent model, making Theorem 1’s assertion about exponential decay rather redundant.

- Theory 2 seems unnecessary: It’s clear that concatenating multiple weight matrices with LoRA applied to the larger matrix creates dependencies. However, there's no practical advantage demonstrated here, nor is there an empirical basis for this approach.

- Line 274: Why use MMLU as a proxy metric? For models at this scale, MMLU scores are generally around random (i.e., ~25%), leading to limited interpretability. Additionally, only the differential score is provided, without the original MMLU results, which weakens MMLU’s reliability as a metric in this context.

- Instruction tuning experiments: Pythia feels like a weak baseline. And is it plausible to claim Mamba has an “emergent ability” (line 439) for in-context learning after instruction tuning, given SSMs' inherent limitations, as noted in [1,2]? Figure 3 also shows a performance drop for Mamba2 as shot count rises, likely due to state-space models’ well-known difficulties with long-sequence processing and in-context learning.

[1] https://arxiv.org/abs/2402.18510

[2] https://arxiv.org/abs/2402.18668

**Questions:**

See weakness.

---

> ### Author Response · Authors · 2024-11-13
> **Reply to comments on theoretical results**
>
> We thank the reviewer for their time and feedback.  Please find our response to your comments below.
>
> > Line 51: For some training framework it does not keep master weight in fp32 which problematic for Mamba. But Mamba was still trained with mixed precision using AMP.
>
> We respectfully point out a potential misunderstanding.  Lines 48-53 are stating official Mamba sources suggest model training be performed in full precision, i.e., fp32.  The source of this concern is that recurrent neural models are prone to instability, discussed at length in [1, 2].
>
> The main point of the paper is to theoretically show that, unlike previously released recurrent models, Mamba SSMs are guaranteed to to be stable (Theorems 1 and 2).  Leveraging this theory, we then show that it is possible to efficiently train Mamba SSMs using mixed-precision, i.e., AMP.  Finally, we note that the paper exhaustively trains Mamba SSMs in full precision (without AMP), and compares learning using AMP to the full precision models (i.e., Figures 1 and 2, and Table 1).
>
> > Theory 1 seems shallow: Isn't Lemma 1 trivial to derive? ( dx_{t+1}/dx_t ) directly represents the decay rate, which must be (\leq 1) to prevent exponential growth in cumulative products. This insight is obvious for any recurrent model, making Theorem 1’s assertion about exponential decay rather redundant.
>
> We respectfully disagree.  Firstly, as previously noted, the stability of recurrent neural models is not a given, with prior works promoting adjustment of the training schedule to combat unstable dynamics of such recurrent models [1, 2].  Secondly, calculating the lypanuv exponents of a recurrent model is nontrivial, i.e., we refer the reviewer to the proof of Lemma 1, which requires careful analysis of Mamba's state-space equations, variable activations, and variable domains.
>
> > Theory 2 seems unnecessary: It’s clear that concatenating multiple weight matrices with LoRA applied to the larger matrix creates dependencies. However, there's no practical advantage demonstrated here, nor is there an empirical basis for this approach.
>
> We respectfully disagree; given an arbitrary linear layer in a network, adding LoRA adapters to such a linear layer's weight matrix does not necessarily induce dependencies *across different model variables*.  As an example, rather than the large global memory cache currently used, assume Mamba was designed with individual linear-layer-weight matrices for the B_t, C_t, and \Delta_t variables.  Clearly, adding LoRA adapters to each individual linear layer would not cause any weight tying for this design choice.  In contrast, the goal of Theorem 3 is to shed light on how the actual design of the recently released Mamba SSMs affects the most widely used PEFT method, LoRA.
>
> We are currently adding empirical results to demonstrate the importance of Theorem 3, and will comment with the updated experiments.
>
> ## References
> [1] Pascanu, R. "On the difficulty of training recurrent neural networks." arXiv preprint arXiv:1211.5063 (2013).
>
> [2] Mikhaeil, Jonas, Zahra Monfared, and Daniel Durstewitz. "On the difficulty of learning chaotic dynamics with RNNs." Advances in Neural Information Processing Systems 35 (2022): 11297-11312.

---

> > ### Author Response · Authors · 2024-11-13
> > **Reply to comment on instruction tuning experiments**
> >
> > We thank the reviewer for their comments regarding our experiments, and address them below.
> >
> > >  Line 274: Why use MMLU as a proxy metric? For models at this scale, MMLU scores are generally around random (i.e., ~25%), leading to limited interpretability. Additionally, only the differential score is provided, without the original MMLU results, which weakens MMLU’s reliability as a metric in this context.
> >
> > This is standard practice, please see [3].
> >
> > > Instruction tuning experiments: Pythia feels like a weak baseline. And is it plausible to claim Mamba has an “emergent ability” (line 439) for in-context learning after instruction tuning, given SSMs' inherent limitations, as noted in [1,2]? Figure 3 also shows a performance drop for Mamba2 as shot count rises, likely due to state-space models’ well-known difficulties with long-sequence processing and in-context learning.
> >
> > As noted on lines 282-283, **Pythia and Mamba models were both pretrained using the same corpus, allowing the fairest comparison between SSMs and Transformers.**.
> >
> > From [4], we state the definition of an emergent ability:
> >
> > *An ability is emergent if it is not present in smaller models but is present in larger models.*
> >
> > Thus, for model families above a particular size, consistently achieving non-negative performance in Figure 4 equates to an emergent ability.  Hence, both Mamba and Mamba-2 instruction tuned parameters conistently display ICL as an emergent ability for parameter counts of 370 million and greater.
> >
> > With regards to SSMs' inherent limitations [1,2], we note that the cited references discuss such limitations for associative recall, a synthetic task.  However, [5] has already shown that Mamba models with a larger recurrent state (e.g., Mamba-2 models) can solve this synthetic task exactly.  Furthermore, *we novelly study ICL on natural language tasks*, and show that instruction tuned Mamba models demonstrate strong ICL relative to comparable Transformers on such tasks.  Thus, we have shown that Mamba instruction-tuned CL performance is not inherently limited for natural language tasks.
> >
> > ## References
> >
> > [1] https://arxiv.org/abs/2402.18510
> >
> > [2] https://arxiv.org/abs/2402.18668
> >
> > [3] Dettmers, Tim, et al. "QLoRA: efficient finetuning of quantized LLMs (2023)." arXiv preprint arXiv:2305.14314 52 (2023): 3982-3992.
> >
> > [4] Wei, Jason, et al. "Emergent abilities of large language models." arXiv preprint arXiv:2206.07682 (2022).
> >
> > [5] Dao, Tri, and Albert Gu. "Transformers are SSMs: Generalized models and efficient algorithms through structured state space duality." arXiv preprint arXiv:2405.21060 (2024).

---

> > > ### Author Response · Authors · 2024-11-23
> > > **New results demonstrating Theorem 3 and follow up**
> > >
> > > Once again, we thank the reviewer for reviewing our submission.  We have added new experiments (Section 4.2 of the new submission draft) demonstrating the implications of Theorem 3 (i.e., parameter tying of time-varying parameters via LoRA's target layer).  The new results demonstrate that targeting the weight parameter described by Theorem 3 near uniformly results in better performance than only targeting other layers, with the former outperforming the latter on 32 out of 35 natural language tasks.  Additional experiments show that *only targeting this weight parameter outperforms targeting all linear layers using LoRA on a majority of tasks*, thus demonstrating a tradeoff between adapting only regularized parameters (i.e., those described by Theorem 3) versus adapting more parameters (a subset of which are unregularized).  We refer the reviewer to the updated submission for further details.
> > >
> > > Respectfully, we also seek to follow up on our earlier comments clarifying the reviewer's comments on our theoretical work.  As noted by the other reviewers, our theoretical results are novel and strong contributions to the growing study of Mamba LLMs.
> > >  If there are further questions we can clarify, we look forward to discussing.

---

> > > ### Comment · Reviewer_jrwp · 2024-11-25
> > >
> > > >    This is standard practice, please see [3].
> > >
> > > Regarding MMLU, it is a standard practice only when the model is large enough to produce meaningful results. I don’t understand how these small-scale models could yield meaningful results. Can you provide the specific MMLU score to verify if it is significantly beyond 25%? A reminder here: 25% is the random accuracy for MMLU. **Relative improvement on this task is meaningless, as all models perform equally poorly, producing random-level accuracy.**
> > >
> > > > Pythia and Mamba models were both pretrained using the same corpus, allowing the fairest comparison between SSMs and Transformers..
> > >
> > > Regarding the second point, although Mamba and Pythia use the same corpus, Pythia's training recipe is notably suboptimal. Using a LLaMA-style training recipe would undoubtedly yield better results than Mamba (anyone experienced in training these models can confirm this). **The baselines really need to be improved.**
> > >
> > > >  However, [5] has already shown that Mamba models with a larger recurrent state (e.g., Mamba-2 models) can solve this synthetic task exactly.
> > >
> > > From what I observe in [5], Mamba2 still struggles with the MQAR task as the number of KV pairs and sequence length increase. Where do you see evidence that Mamba2 can solve these tasks *exactly*? The paper I cited highlights fundamental limitations of recurrent models, which cannot be resolved simply by expanding the state size. Therefore, I find it quite inappropriate to claim "emergent ability in in-context learning" here.

---

> ### Comment · Reviewer_jrwp · 2024-11-25
>
> > We respectfully point out a potential misunderstanding. Lines 48-53 are stating official Mamba sources suggest model training be performed in full precision, i.e., fp32. The source of this concern is that recurrent neural models are prone to instability, discussed at length in [1, 2].
>
> They explicitly state that Mamba was trained using AMP mixed precision. Why do you keep insisting that Mamba requires pure FP32 precision for training?
>
>  >> Our models were trained using PyTorch AMP for mixed precision. AMP keeps model parameters in float32 and casts to half precision when necessary
>
> For example, linear projection would be cast to BF16 for matmuls, and the resulting BF16 Mamba parameters could then be sent to the Mamba kernel (except for A).
>
>
> > Firstly, as previously noted, the stability of recurrent neural models is not a given, with prior works promoting adjustment of the training schedule to combat unstable dynamics of such recurrent models [1, 2].
>
> A recent work (https://arxiv.org/abs/2303.06349) shows that stability can be achieved simply by regularizing the eigenvalue norm of the decay to be smaller than 1. Also, given that all recent RNN decays are already smaller than 1 and inherently stable, what is the purpose of proving this again?
>
> > adding LoRA adapters to each individual linear layer would not cause any weight tying for this design choice. In contrast, the goal of Theorem 3 is to shed light on how the actual design of the recently released Mamba SSMs affects the most widely used PEFT method, LoRA.
>
> I still do not see the motivation for discussing the application of LoRA on a larger concatenated projection as opposed to using individual LoRAs on different projection components.
>
> > We are currently adding empirical results to demonstrate the importance of Theorem 3, and will comment with the updated experiments.
>
> Where are your results?

---

> ### Author Response · Authors · 2024-11-26
> **Code of conduct**
>
> Firstly, we greatly appreciate the reviewer participating in the discussion.  We are very grateful for the reviewer's time and effort, and we also hope that the reviewer is respectful of the time and work required both for our submission and the additional experiments/responses crafted during the rebuttal period.
>
> Secondly, we politely request the reviewer engage in a respectful discussion.  The following language from previous replies are disrespectful and, at times, disparaging/accusatory:
> - "Why do you keep insisting "
> - "Nothing special."
> - "If all recent rnn's decay are smaller than 1 and already stable, what's the point of proving it again?"
> - "Where do you see evidence that Mamba2 can solve these tasks exactly?"
> - "would undoubtedly yield better results than Mamba (anyone experienced in training these models can confirm this)"
>
> For reference, the following is the ICLR code of ethics, which all conference participants are meant to abide by:
> https://iclr.cc/Conferences/2025/CodeOfConduct
>
> # Questions have been answered in previous author replies
> As previously stated, we are very appreciative of the reviewer's time. With all due respect and awareness of both the reviewer's and authors' time, we kindly ask that responses be carefully read.  E.g.:
> >> We are currently adding empirical results to demonstrate the importance of Theorem 3, and will comment with the updated experiments.
>
> > Where are your results?
>
> The answer to the reviewer's question directly appears in the preceding author response, aptly titled **New results demonstrating Theorem 3 and follow up**
>
> This is also true for the following statement:
> > From what I observe in [5], Mamba2 still struggles with the MQAR task as the number of KV pairs and sequence length increase. Where do you see evidence that Mamba2 can solve these tasks exactly? The paper I cited highlights fundamental limitations of recurrent models, which cannot be resolved simply by expanding the state size. Therefore, I find it quite inappropriate to claim "emergent ability in in-context learning" here.
>
> The following is the arxiv link for [5]:
> https://arxiv.org/pdf/2405.21060
>
> **In Figure 8 of the above (page 28), Mamba-2 very clearly solves MQAR exactly for model dimensions 128 and greater**.
>  Furthermore, Mamba-2 with dimension 256 outperforms the evaluated Llama-equivalent architecture (called "Attention" in the plots) as well as the recently released linear-attention-with-sliding-window architecture (called "Based", which was designed with MQAR in mind).  [5] also states: "the larger state allows more information (key-value pairs) to be memorized."
>
> Finally, this is also true in the following:
> > A recent work (https://arxiv.org/abs/2303.06349) shows that stability can be achieved simply by regularizing the eigenvalue norm of the decay to be smaller than 1. Also, given that all recent RNN decays are already smaller than 1 and inherently stable, what is the purpose of proving this again?
>
> We thank the reviewer for supplying the source of their criticism.  To be clear, the RNNs discussed in (https://arxiv.org/abs/2303.06349) contain **time-invariant** matrices A,B.  **This result does not extend to time-varying matrices A_t, B_t, such as those used in Mamba SSMs**.  This point is raised in our submission, lines 220-228, where we discuss related previous work and the stability results of S4 models (which the reviewer-supplied paper "Resurrecting Recurrent Neural Networks for Long Sequences" was written in response to, and which preceed Mamba SSMs).
>
> # Claims
> Finally, we also request that important claims be backed up with evidence, e.g.:
> > Using a LLaMA-style training recipe would undoubtedly yield better results than Mamba (anyone experienced in training these models can confirm this)
>
> We respectfully ask that the user supply a reference which tests the stated hypothesis and directly supports the stated conclusion.  The ICLR reviewer guidelines states reviewers should "provide the additional evidence" for their assessment.
>
> # Respectful discussion proceeding
> Again, we are appreciative of the time required to carefully review and critique our submission.  We are eager to discuss the reviewer's remaining comments and politely request the discussion remain respectful going forward.

---

> > ### Comment · Reviewer_jrwp · 2024-11-26
> >
> > Dear Author, can you comment on MMLU score first?

---

> > > ### Author Response · Authors · 2024-11-26
> > > **MMLU**
> > >
> > > Dear Reviewer jrwp,
> > >
> > > Thank you for your reply.  Of course, as previously noted, we are eager to address all reviewer criticisms and, as a reminder, request the discussion remain respectful going forward.
> > >
> > > > Regarding MMLU, it is a standard practice only when the model is large enough to produce meaningful results. I don’t understand how these small-scale models could yield meaningful results. Can you provide the specific MMLU score to verify if it is significantly beyond 25%? A reminder here: 25% is the random accuracy for MMLU. Relative improvement on this task is meaningless, as all models perform equally poorly, producing random-level accuracy.
> > >
> > > Firstly, we note that **MMLU was only used in Table 1 and Figure 1 to demonstrate divergence under AMP/MPFT/PEFT**.  Irregardless of the actual performance of the model, the following is true: using the Eleuther eval harness, running the same model with the same settings will produce the same exact results on the MMLU dataset (and any dataset).  Thus, if we evaluate a model trained with (full fine-tuning, fp32) and a model trained with (full fine-tuning, bf16) and there is a large discrepancy in performance, this does not speak to the model's overall ability to solve the task, but to deviations in the final learned parameter (due to mixed-precision, compounded by training iterations).  Clearly, all the settings remain the same, except the two models; if we make both models the same (i.e., feed the same checkpoint for both evaluations), divergence would be zero.
> > >
> > > Secondly, thanks to the helpful suggestions from Reviewer tWTf, our divergence results also include evaluation using Winogrande, and additional fine-tuning datasets (LIMA).  Found in Appendix I, these additional results once again show Mamba LLMs are drastically more stable than Transformer-based LLMs.
> > >
> > > We greatly appreciate the reviewers diligence on the topic of MMLU performance.  We fully agree with the comment regarding model's of this size and MMLU; if we were using MMLU to actually grade the various model's instruction-tuning/ICL performance, this would not accurately reflect practical solving of this task unless there was a distinct increase over 25%.  Thus, we note that we have only used this dataset to grade divergence, and not the instruction-tuned performance of any of the considered models.
> > >
> > > For reference, here are the raw LIMA MMLU numbers for full fine-tuning:
> > >
> > > Architecture and precision | Size | 0-shot | 1-shot | 3-shot | 5-shot|
> > > -------------------------------|------|---------|---------|---------|-------|
> > > Mamba, fp32 | 130M | 22.76 | 24.76 | 25.21 | 24.36 |
> > > Mamba, fp16 | 130M | 22.70 | 24.70 | 25.18 | 24.35 |
> > > Mamba, bf16 | 130M | 22.78 | 24.85 | 25.24 | 24.43 |
> > > Pythia, fp32 | 160M | 22.98 | 25.01 | 26.30 | 26.37 |
> > > Pythia, fp16 | 160M | 23.08 | 25.00 | 26.61 | 25.70 |
> > > Pythia, bf16 | 160M | 23.58 | 25.64 | 26.27 | 24.82 |
> > > OpenELM, fp32 | 270M | 23.52 | 25.47 | 27.48 | 26.24 |
> > > OpenELM, fp16 | 270M | 23.55 | 25.40 | 27.47 | 26.26 |
> > > OpenELM, bf16 | 270M | 23.50 | 25.30 | 27.4 | 26.27 |
> > > Mamba, fp32 | 370M | 24.23 | 25.67 | 25.38 | 25.21 |
> > > Mamba, fp16 | 370M | 24.28 | 25.67 | 25.45 | 25.31 |
> > > Mamba, bf16 | 370M | 24.08 | 25.56 | 25.43 | 25.27 |
> > > Pythia, fp32 | 410M | 26.33 | 26.89 | 26.88 | 25.34 |
> > > Pythia, fp16 | 410M | 25.87 | 27.10 | 26.97 | 27.06 |
> > > Pythia, bf16 | 410M | 25.32 | 26.66 | 27.50 | 26.68 |
> > > OpenELM, fp32 | 450M | 25.26 | 25.63 | 25.79 | 26.29 |
> > > OpenELM, fp16 | 450M | 25.17 | 25.47 | 25.62 | 26.26 |
> > > OpenELM, bf16 | 450M | 24.73 | 25.05 | 25.73 | 25.15 |
> > > Mamba, fp32 | 790M | 23.97 | 24.84 | 25.55 | 25.13 |
> > > Mamba, fp16 | 790M | 23.96 | 24.94 | 25.51| 25.15 |
> > > Mamba, bf16 | 790M | 23.89 | 24.98 | 25.50 | 25.10 |
> > > Pythia, fp32 | 1B | 22.83 | 24.43 | 24.56 | 25.42 |
> > > Pythia, fp16 | 1B | 22.85 | 24.43 | 24.53 | 25.43 |
> > > Pythia, bf16 | 1B | 22.91 | 24.31 | 24.68 | 25.48 |
> > >
> > > Mamba does well at this stability stress test, while OpenELM does slightly less well, and Pythia is far less stable.  To further test stability, Appendix I includes the following configurations for stability testing: (Alpaca, Winogrande), (Alpaca, MMLU), (LIMA, Winogrande), (LIMA, MMLU)

---

> > > > ### Author Response · Authors · 2024-11-26
> > > > **Acknowledgement of "Code of conduct" comment**
> > > >
> > > > Dear Reviewer jrwp,
> > > >
> > > > We do not want to draw attention away from our comment addressing your MMLU conerns ([here](https://openreview.net/forum?id=i9RTCC6whL&noteId=srBjUZwrRf)), but can the reviewer kindly acknowledge our earlier comment, titled "[Code of conduct](https://openreview.net/forum?id=i9RTCC6whL&noteId=srBjUZwrRf)?"
> > > >
> > > > In particular, can the reviewer kindly acknowledge the following points in the reiterated section below?  Please note that in the final point, **regarding the reviewer-supplied paper "Resurrecting Recurrent Neural Networks for Long Sequences," the results of that paper do not extend to Mamba SSMs and, thus, do not relate to our theoretical results**.  Thus, this directly impacts the reviewer's earlier comment that our theoretical results are trivial.  We maintain that our theoretical results are nontrivial and novel.
> > > >
> > > > # Questions have been answered in previous author replies
> > > > As previously stated, we are very appreciative of the reviewer's time. With all due respect and awareness of both the reviewer's and authors' time, we kindly ask that responses be carefully read.  E.g.:
> > > > >> We are currently adding empirical results to demonstrate the importance of Theorem 3, and will comment with the updated experiments.
> > > >
> > > > > Where are your results?
> > > >
> > > > The answer to the reviewer's question directly appears in the preceding author response, aptly titled **New results demonstrating Theorem 3 and follow up**
> > > >
> > > > This is also true for the following statement:
> > > > > From what I observe in [5], Mamba2 still struggles with the MQAR task as the number of KV pairs and sequence length increase. Where do you see evidence that Mamba2 can solve these tasks exactly? The paper I cited highlights fundamental limitations of recurrent models, which cannot be resolved simply by expanding the state size. Therefore, I find it quite inappropriate to claim "emergent ability in in-context learning" here.
> > > >
> > > > The following is the arxiv link for [5]:
> > > > https://arxiv.org/pdf/2405.21060
> > > >
> > > > In Figure 8 of the above (page 28), Mamba-2 very clearly solves MQAR exactly for model dimensions 128 and greater.
> > > >  Furthermore, Mamba-2 with dimension 256 outperforms the evaluated Llama-equivalent architecture (called "Attention" in the plots) as well as the recently released linear-attention-with-sliding-window architecture (called "Based", which was designed with MQAR in mind).  [5] also states: "the larger state allows more information (key-value pairs) to be memorized."
> > > >
> > > > Finally, this is also true in the following:
> > > > > A recent work (https://arxiv.org/abs/2303.06349) shows that stability can be achieved simply by regularizing the eigenvalue norm of the decay to be smaller than 1. Also, given that all recent RNN decays are already smaller than 1 and inherently stable, what is the purpose of proving this again?
> > > >
> > > > We thank the reviewer for supplying the source of their criticism.  To be clear, the RNNs discussed in (https://arxiv.org/abs/2303.06349) contain **time-invariant** matrices A,B.  **This result does not extend to time-varying matrices A_t, B_t, such as those used in Mamba SSMs**.  This point is raised in our submission, lines 220-228, where we discuss related previous work and the stability results of S4 models (which the reviewer-supplied paper "Resurrecting Recurrent Neural Networks for Long Sequences" was written in response to, and which preceed Mamba SSMs).

---

> ### Comment · Reviewer_jrwp · 2024-11-27
>
> Thank you for your clarifications.
>
> > MMLU: We do not want to draw attention away from our comment addressing your MMLU conerns
>
> Thank you for finally providing the MMLU scores. As the authors themselves acknowledge, the MMLU scores are near random and therefore meaningless. Given this, Table 1, Figure 1, and Table 4 are highly misleading and should be removed from the paper immediately.
>
> >  Code of conduct
>
> I apologize for this. Could you also address my other question? For instance, do you agree that Pythia is a weak baseline?
>
> > Mamba2 MQAR
>
> Do you agree with this: as the sequence length and the number of key-value (KV) pairs increase, Mamba2 will no longer solve this task exactly? Could you comment on the paper I cited, which reveals the fundamental limitations of recurrent models in in-context learning? Do you find the claim regarding the *emergence of in-context learning ability* both proper and rigorous?
>
> > regarding the reviewer-supplied paper "Resurrecting Recurrent Neural Networks for Long Sequences," the results of that paper do not extend to Mamba SSMs and, thus, do not relate to our theoretical results
>
> I respectfully disagree. Linear RNN can be unfolded to obtain "attention" matrix with attention entries being cumulative product of decay from j-th position to i-th position. The only requirement is that decay should be smaller than 1, so the cumulative product won't explode, regardless of whether decay is data-dependent or not. Your proof heavily reply on the fact that $(dx_{t+1}/dx_t < 1)$, which is equivalent to decay < 1, as I mentioned. Mamba is a special case of linear RNN, so it should be applicable to Mamba as well. Do you agree with this? If not, please specify reasons. If so, I find the proof's theoretical significance less valuable, which hinders this work's contribution.

---

> ### Author Response · Authors · 2024-11-27
> **Not an appropriate takeaway for MMLU results**
>
> Dear Reviewer jrwp,
>
> We sincerely thank you for apologizing for your earlier breach of conduct, we greatly appreciate it.
>
> With regards to MMLU, the following:
> > Given this, Table 1, Figure 1, and Table 4 are highly misleading and should be removed from the paper immediately.
>
> ignores the entire discussion provided in our earlier comment and is not an appropriate conclusion.  Very respectfully, as mentioned in our earlier comment [here](https://openreview.net/forum?id=i9RTCC6whL&noteId=AktTlGzfDI), we kindly ask that responses be carefully read.  For convenience, the discussion covering divergence may be found [here](https://openreview.net/forum?id=i9RTCC6whL&noteId=srBjUZwrRf).
>
> # Evidence-based criteria
>
> We also kindly ask the reviewer to refrain from judging our submission based on hypotheticals, to judge the contributions within scope, and to keep the discussion evidence-based.  For instance:
> > Do you agree with this: as the sequence length and the number of key-value (KV) pairs increase, Mamba2 will no longer solve this task exactly? Could you comment on the paper I cited, which reveals the fundamental limitations of recurrent models in in-context learning? Do you find the claim regarding the emergence of in-context learning ability both proper and rigorous?
>
> To be very clear, Mamba models exhibit ICL.  This has already been tackled in earlier work [1], our contribution is to assess this for natural language tasks.  Again, as shown in Figure 3 of the submission, Mamba models exhibit ICL on natural language tasks.  Yes, we consider this both proper and rigorous: emergent abilities are empirically demonstrated (please see [2]), and this property was empirically demonstrated in Figure 4.
>
> > Could you also address my other question? For instance, do you agree that Pythia is a weak baseline?
>
> As detailed in the official ICLR reviewer guide and discussed in the "Claims" section in our comment [here](https://openreview.net/forum?id=i9RTCC6whL&noteId=AktTlGzfDI), we ask the reviewer to please keep criticisms evidence based.  Initially, the reviewer criticism was "Pythia feels like a weak baseline."   We respectfully list reasons why Pythia is used, and why it is an appropriate baseline:
> 1. published in a reputable venue (ICML '23)
> 2. same training data as Mamba/Mamba-2
> 3. same number of pretraining steps as Mamba/Mamba-2
> 4. range of increasing model sizes
>
> As described in [2], (2), (3), and (4) are particularly important parameters for LLM emergent abilities, and thus comparing models with these properties is important for an apples-to-apples comparison of emergent abilities.  Please note that we also test other Transformer-based baselines (i.e., OLMo, TinyLlama, and OpenELM), please see Figure 4.  These other Transformer LLMs were trained on larger datasets (i.e., a superset of the Pythia/Mamba training data) and for longer pretraining steps; thus, comparisons between Pythia/Mamba/Mamba-2 and these other models are actually biased in the other models' favor.
>
> To reiterate, we kindly ask the reviewer to read lines 415-423 of the submission, which state "that direct comparisons between Mamba, Mamba-2, and Pythia are the most fair (as these three classes of models were all pretrained on the same dataset for the same number of total pretraining tokens)."  Thus, Pythia is an appropriate baseline.
>
> # References
> [1] Park, Jongho, et al. "Can mamba learn how to learn? a comparative study on in-context learning tasks." arXiv preprint arXiv:2402.04248 (2024).
>
> [2] Wei, Jason, et al. "Emergent abilities of large language models." arXiv preprint arXiv:2206.07682 (2022).

---

> > ### Author Response · Authors · 2024-11-27
> > **Mamba is not a special case of linear RNNs**
> >
> > Dear Reviewer jrwp,
> >
> > We greatly appreciate the reviewer seeking clarification, and thank them again for participating in the discussion.  The following is the reviewer's comment regarding Mamba models being equivalent to linear RNNs:
> >
> > > I respectfully disagree. Linear RNN can be unfolded to obtain "attention" matrix with attention entries being cumulative product of decay from j-th position to i-th position. The only requirement is that decay should be smaller than 1, so the cumulative product won't explode, regardless of whether decay is data-dependent or not. Your proof heavily reply on the fact that
> > , which is equivalent to decay < 1, as I mentioned. Mamba is a special case of linear RNN, so it should be applicable to Mamba as well. Do you agree with this? If not, please specify reasons. If so, I find the proof's theoretical significance less valuable, which hinders this work's contribution.
> >
> > There are several leaps in logic in the above, and, in particular, the following statement is incorrect: "Mamba is a special case of linear RNN."
> >
> > Please allow us to go through this very slowly, without any jumping to conclusions.  From [1], the following are the state equations for linear RNNs:
> >
> > $x_k = A x_{k-1} + B u_k, y_k = C x_k + D u_k$
> >
> > The following are the state equations for Mamba SSMs:
> >
> > $x_k = A_k x_{k-1} + B_k u_k, y_k = C_k x_k$
> >
> > Notice, the matrices in the linear RNN's state equations are time-invariant whereas the matrices in the Mamba state equations are time-varying, i.e., $A, B, C, D$ versus $A_k, B_k, C_k$.
> >
> > Clearly, if $A_k, B_k, C_k$ vary for values of $k$, the Mamba state-space equations cannot be represented by the linear RNNs state equations.  **Thus, Mamba is not a special case of linear RNNs.**
> >
> > Again, we sincerely appreciate the reviewer seeking clarification and engaging in the discussion.
> >
> > # References
> > [1] Orvieto, Antonio, et al. "Resurrecting recurrent neural networks for long sequences." International Conference on Machine Learning. PMLR, 2023.

---

> ### Comment · Reviewer_jrwp · 2024-11-27
>
> Dear Authors,
>
> Linear RNNs are characterized by their linear recurrence, which allows for parallelization. Could you clarify why you maintain that Mamba does not fall under the category of linear RNNs? Wouldn't a gated linear RNN also be considered a linear RNN?
>
> Additionally, could you comment on the first sentence of the abstract from the paper co-authored by Tri Dao, titled *"The Mamba in the Llama: Distilling and Accelerating Hybrid Models"* (https://arxiv.org/abs/2408.15237):
>
> > "Linear RNN architectures, like Mamba, can be competitive with Transformer models in language modeling while having advantageous deployment characteristics..."
>
> Do you believe Tri Dao, one of the authors of the remarkable Mamba paper, would agree with your statement that Mamba is not a special case of linear RNNs? I look forward to your response.

---

> > ### Author Response · Authors · 2024-11-27
> > **Please view earlier comment**
> >
> > Dear Reviewer jrwp,
> >
> > Please view the earlier comment [here](https://openreview.net/forum?id=i9RTCC6whL&noteId=8xMPEMQwb5).  Again, we please ask to go through this very slowly, without any jumping to conclusions.
> >
> > Firstly, you have cited [1] as the source for the triviality of our theoretical results.  We want to make this extremely clear; [1] uses the term "Linear RNNs" to describe RNNs governed by the following state equations:
> >
> > $x_k = A x_{k-1} + B u_k, y_k = C x_k + D u_k$
> >
> > Thus, the state equations for "Linear RNNs" are *linear, time-invariant*.
> >
> > Let us look once more at the Mamba state equations:
> >
> > $x_k = A_k x_{k-1} + B_k u_k, y_k = C_k x_k$
> >
> > The mamba state equations are *linear, time-varying.*
> >
> > **Firstly, we very respectfully note that the stability work the reviewer has cited to claim our theoretical work is trivial does not deal with time-varying (i.e., Mamba) parameters.**  Please, we very respectfully ask for a moment to carefully consider the last statement.
> >
> > For your follow up comment, please notice that for both sets of state equations above, they are described as *linear*.  Now, very carefully, Mamba can be described as a linear RNN because its state space equations are *linear, time-varying,* but this is not the "Linear RNN" from [1] whose state space equations are "linear, time-invariant."
> >
> > Again, we are very appreciative of the reviewers time and willingness to discuss.  We very politely ask again that the reviewer please carefully go over our earlier comment [here](https://openreview.net/forum?id=i9RTCC6whL&noteId=8xMPEMQwb5).
> >
> > # References
> > [1] Orvieto, Antonio, et al. "Resurrecting recurrent neural networks for long sequences." International Conference on Machine Learning. PMLR, 2023.

---

> > > ### Comment · Reviewer_jrwp · 2024-11-27
> > >
> > > Linear RNNs clearly have a broader definition. Do you agree that their remarks on bounding eigenvalues to less than 1 for stabilizing training also apply to gated linear RNNs with time-varying systems? It should be noted that all gated linear RNN models (e.g., Mamba, Griffin, RWKV6, GLA, HGRN, HGRN2) enforce gating values smaller than 1 for this very reason. Does your proof also rely on this assumption? Isn’t this widely acknowledged in the community?

---

> ### Comment · Reviewer_jrwp · 2024-11-27
>
> Regarding MMLU explanations: If a model performs at random-level accuracy, failing entirely in this task, the stability analyses seem meaningless to me. You still have ample rebuttal time to present stability analyses on other tasks.
>
> Regarding Pythia: Pythia employs a Transformer architecture that is rarely used today. [This study](https://openreview.net/pdf?id=i8tGb1ab1j) also reports that Pythia is sensitive to bit-precision, unlike other Transformer architectures that may not necessarily exhibit such sensitivity. Please test on a more recent and well-trained Llama-based architecture, such as the Qwen series.

---

> ### Author Response · Authors · 2024-11-27
> **Inappropriate request for experiments**
>
> Dear Reviewer jrwp,
>
> > Regarding MMLU explanations: If a model performs at random-level accuracy, failing entirely in this task, the stability analyses seem meaningless to me. You still have ample rebuttal time to present stability analyses on other tasks.
>
> Again, we thank the reviewer for engaging in the discussion.  Please note, per the reviewer guidelines sent for the discussion period extension (now), reviewers have been instructed to not ask for significant experiments.  Each stability experiment (e.g., Figure 1) requires fine-tuning 72 models and 288 few-shot evaluations; the request that we perform more of these experiments is inappropriate.
>
> Respectfully, as we have requested multiple times, we ask that the reviewer please carefully read previous replies.  As mentioned in our earlier [comment](https://openreview.net/forum?id=i9RTCC6whL&noteId=srBjUZwrRf), we have already supplied additional stability results using other benchmarks and fine-tuning datasets.  We also respectfully ask the reviewer to read the previous discussion on MMLU.
>
> > Regarding Pythia: Pythia employs a Transformer architecture that is rarely used today. This study also reports that Pythia is sensitive to bit-precision, unlike other Transformer architectures that may not necessarily exhibit such sensitivity. Please test on a more recent and well-trained Llama-based architecture, such as the Qwen series.
>
> Again, a request for further instruction tuning ICL experiments is inappropriate.  Also, as previously mentioned above, we please ask that the reviewer carefully read author comments; as mentioned in our reply to the reviewer earlier ([here](https://openreview.net/forum?id=i9RTCC6whL&noteId=OCYy3pczoU)), we have already evaluated several other recent Transformer-based models (OLMo, TinyLlama, and OpenELM).
>
> The report the reviewer linked shows Pythia is sensitive to 3-bit precision, and stable for 4-bit precision and higher.  Our experiments never consider precisions lower than 16-bit.

---

> ### Comment · Reviewer_jrwp · 2024-11-27
>
> Let’s be direct. I could simply choose not to respond, rate this submission poorly, and likely see it rejected. However, I want to share some thoughts below. In my view, this work makes an ambiguous contribution and falls short of meeting the standards of a prestigious venue like ICLR.
>
> First, the stability analysis builds on well-known facts about RNN models, specifically that their decay terms are bounded by 1. Theorems 1 and 2 lack technical significance; they essentially restate the intuitive property that the exponential decay in linear RNNs reduces the impact of earlier deviations on later positions. This is neither surprising nor particularly insightful. My takeaway is that the authors attempt to use the concept of Lyapunov stability to highlight their theoretical contribution; however, the theoretical analysis strikes me as somewhat trivial.
>
> Second, I am not convinced that numerical stability is a significant issue to address here. The Mamba model was trained with bf16 mixed precision without notable difficulty (To my knowledge, only matrix \( A \) needs to be stored in fp32, while all other linear projection parameters can safely use bf16—a method I’ve applied in practice without issues. I believe Mamba's official repository suggests is for the same reasons). Since mixed precision is not a real practical challenge during training, using it for fine-tuning is simply straightforward. In summary, my question is: why should we focus on numerical stability for fine-tuning when training the model with mixed precision does not present any much issues?
>
> The application of LoRA and instruction tuning to Mamba is similarly unremarkable. These techniques are easily adaptable to new architectures. It feels like low-hanging fruit that lacks novelty. For instance, [this paper](https://arxiv.org/abs/2411.03855) executes PEFT for Mamba with greater polish, despite the authors might argue that it was concurrent work released after the ICLR submission deadline.
>
> Finally, the claim about unlocking in-context learning ability for Mamba through instruction tuning as a core contribution fails to convince me. As you mentioned, Mamba has already been shown to exhibit some degree of in-context learning ability, but that is the contribution of other work. This paper merely adds some limited real-world task evaluation. Moreover, instruction tuning has been widely used for a long time, as demonstrated by models available at [Hugging Face](https://huggingface.co/models?search=mamba%20instruct).

---

### Official Review · Reviewer_tWTf · 2024-11-04

**Soundness:** 4
**Presentation:** 3
**Contribution:** 3
**Rating:** 6
**Confidence:** 4

**Summary:**

- The authors work to demonstrate that the Mamba SSM is robust to small input perturbations during training, meaning that Mamba is compatible with mixed-precision fine-tuning (MPFT) and parameter-efficient fine-tuning (PEFT). The authors include both theoretical and empirical evidence for this finding.
- Theory results are in section 3, using dynamical systems theory (Lyapunov exponents) to demonstrate that Mamba’s recurrent dynamics are robust to small input changes
- Section 5’s empirical results contain the following experiments (this list defines the numbers which I’ll use to refer to experiments later):
  1. Main results: divergence in accuracy between FP32 and FP16 or BF16 inference, showing that the accuracy divergence for Mamba is roughly similar to that for pythia and OLMo
  2. Non-divergent mamba fine-tuning: divergence in MMLU accuracy compared to full-precision for FP16 or BF16 models fine-tuned fully, or using LoRA on all or a subset of linear layers. Shows that divergence for Mamba is generally less than similar-size transformers.
  3. Hardware throughput and memory-utilization improvements: Average tokens-per-second and maximum memory-per-token for mamba of varying sizes, using FP32 full fine-tuning, or FP16/BF16 LoRA on all or a subset of linear layers. Shows that any form of PEFT is faster and less memory-intensive than the FP32 full fine-tuning
  4. Instruction tuning’s impact on Mamba ICL for natural language tasks: authors claim that instruction tuning narrows the ICL gap between Mamba and Pythia
  5. ICL as an emergent ability of Mamba SSMs: compares ICL performance for Mambas and several transformer models of varying sizes

**Strengths:**

- The paper addresses a very important problem that, until now, has helped to prevent the wider use of SSMs: training resources for an SSM have been greater than a similarly-sized transformer, because the transformer has been known to be compatible with efficient fine-tuning techniques. Addressing this disparity will help more people to consider working with SSMs, as their computational requirements will suddenly be within reach
- The combination of both theory and empirical results is quite persuasive and is convincing that Mamba is indeed compatible with MPFT/PEFT
- Theoretical results are simple, elegant and important. The proofs in the appendix are easy to follow and look solid.
- Care has been put into the experimental setup, for instance everything being trained using the same batch size in the main results

**Weaknesses:**

- My most major critique is that there aren’t any error bars or sense of the spread/randomness in the empirical results, especially for experiment 2 / figure 1, experiment 4 / figure 3 for the fine-tuned models and experiment 5 / figure 4 for the fine-tuned models.
-  My second biggest critique is that each experiment is only performed with one dataset (MMLU, fine-tuning with Alpaca for fine-tuning experiments). Including more than one dataset or task would strengthen the claims you’re drawing from the empirical evaluations. Experiment 5 includes more datasets, but the results for all datasets are averaged together and readers can’t consider their performances separately
- The graphs are nice in figures 1-4, but it would also be helpful to include the actual numbers in tables in the appendix
- The paper seems to have a lot of extra contextualization in it, taking up room which could be devoted to more detailed empirical analysis. In particular, section 2 background and section 5 related work seem to just both be disjoint related work sections. Usually I would consider the “background” section to be merely preliminary information and definition of symbols that are necessary to understand the rest of the paper, like a statement of the SSM equations and formal definition of Lyapunov exponent. Then, related work would be a longer section detailing, well, related work. Just as an example for comparison, you can see this differentiation in roles between background and related work in this paper’s sections 2 and 3 here: https://arxiv.org/pdf/2409.00717 Contextualization/related work is important to include, but I would consider prioritizing and moving at least some of it to appendix to include more empirical results
- Some of the number references seem to be wrong:
  - Line 345, “Figure 5” seems to actually be referring to Figure 1
  - Line 379, “Figure 5” seems to refer to Figure 2 (did you maybe put your \label{} command for the figures in the wrong spot inside your figures’ code blocks?)
  - Appendix theorem 3 corresponds to main paper theorem 2.

Overall, I'm setting my rating to 6 due to the significance of the theoretical results. However, I would appreciate more detail to the empirical evaluations as I mention above.

**Questions:**

Why did you choose to work with different sets of transformer models in the main results versus the other experiments? For instance, you give results for OLMo on the first experiment (Table 1) but not second, and results for OpenELM in all experiments but the first one.

---

> ### Author Response · Authors · 2024-11-23
> **New results**
>
> We thank the reviewer for their detailed review and appreciation of our main theoretical results.  Please find our reply to your comments below
>
> # Extra Contextualization
> We appreciate and agree with this feedback.  We've moved the background section to the appendix, which has allowed the inclusion of more detailed empirical analysis in the main text.  In particular, this has allowed ablative experiments demonstrating the impact of Theorem 3 (Section 4.2).
>
> # Each experiment is only performed with one dataset (MMLU, fine-tuning with Alpaca for fine-tuning experiments)
> We completely agree that including more than one dataset or task would strengthen the claims you’re drawing from the empirical evaluations.  To this end, for divergence results, we have included an additional benchmark task (Winogrande) and fine-tuning dataset (LIMA), and repeated the experiment from Figure 1.
>
> These experiments include 72 new fine-tuned LLMs, 360 new MMLU evaluations, and 720 new Winogrande evaluations.  In contrast to OpenELM--which exhibits large deviation spikes for Alpaca fine-tuned models and both benchmark evaluations--and Pythia--which exhibits large deviation spikes for all four fine-tuning dataset and benchmark configurations--**Mamba does not exhibit a single large deviation spike on any benchmark for all considered model sizes and MPFT/PEFT configurations** (i.e., 18 total configurations excluding the full-precision baselines).  Indeed, these additional experiments demonstrate Mamba models are significantly more stable for MPFT/PEFT compared to Transformer-based LLMs. E.g., **for MMLU evaluations, Alpaca fine-tuning with Mamba models is an average 2.6 times smaller in mean divergence than both Pythia and OpenELM models, while LIMA fine-tuning with Mamba models is an average 7 and 3.25 times smaller in mean divergence than Pythia and OpenELM models, respectively**.  All results may be found in Appendix I.
>
> # There aren’t any error bars or sense of the spread/randomness in the empirical results
> We thank the reviewer for raising this point.  For all mean divergence results, we have included the accompanying standard deviation (Appendix I).  We have additionally added the raw numbers for experiment 4 / figure 3 (Appendix F, Tables 5 and 6).  We are adding the remaining tables; if accepted, we will add these to the camera ready.
>
> # Wrong references
> We thank the reviewer for spotting these incorrect references, we have corrected them in the updated manuscript.  We are currently tracking down the source of the remaining incorrect reference.
>
> # Answer to question
> Initially, our plan was to solely compare to Phi in Figure 1's experiments.  However, extensive effort could not get Phi to fine-tune with Alpaca (even with LoRA + mixed-precision) on our available compute.  Thus, due to time and manuscript-space considerations, we decided to diversify the evaluation of Transformer-based OLMo and OpenELM across the two experiments.  We retained Phi in Table 1 to demonstrate instability of Transformer LLMs for one of most popular and performant LLMs of its size.

---

> > ### Comment · Reviewer_tWTf · 2024-11-26
> >
> > Thank you for your explanations and willingness to revise the paper! I especially appreciate the large expansion of the empirical results which allow more confidence to be placed in the results and claims. I will increase my soundness score to 4 for this reason.

---

> > > ### Author Response · Authors · 2024-11-27
> > > **Appreciate the positive feedback and discussion**
> > >
> > > Dear Reviewer tWTf,
> > >
> > > Thank you for the positive feedback and suggestions!  We agree, the large number of additional experiments, along with the accompanying error bars, make the theoretical results much more compelling.  With the additional ablation results to demonstrate Theorem 3, we believe the current paper is much stronger than the original submission (we thank you again for contributing to these changes!).
> > >
> > > Best,
> > >
> > > Authors of Submission 9040

---

> > > > ### Author Response · Authors · 2024-12-02
> > > > **Follow up before the discussion period ends**
> > > >
> > > > Dear Reviewer tWTf,
> > > >
> > > > Touching base before the discussion period closes, please let us know if we've addressed all questions/concerns.  We also wanted to double check whether the reviewer had a chance to view the additional results demonstrating the generalization implications of Theorem 3 (Section 4.2), and whether the reviewer had any questions.
> > > >
> > > > Thanks again for the helpful feedback and positive discussion.
> > > >
> > > > Best,
> > > >
> > > > Authors of Submission 9040

---

### Official Review · Reviewer_Pctm · 2024-11-04

**Soundness:** 3
**Presentation:** 3
**Contribution:** 2
**Rating:** 5
**Confidence:** 3

**Summary:**

This paper aims to address a noticeable gap in research concerning the amenability of the Mamba model to popular fine-tuning frameworks such as PEFT and MPFT. Utilizing theoretical insights from dynamical systems, the paper demonstrates that minor input variations within the SSM layer of either model do not lead to outputs that deviate exponentially. This theoretical assertion is further validated by empirical experiments.

**Strengths:**

1. The paper is well-written and easy to follow.
2. The paper provides a theoretical analysis to support experimental performance.
3. Comprehensive experiments compare the Mamba model with Transformer-based models across different scenarios.

**Weaknesses:**

Though the theoretical analysis of this paper is really solid, since most researchers nowadays use MPFT and PEFT for Mamba, the contribution of this paper may be limited. It may be better if some weaknesses about MPFT and PEFT for Mamba are figured out and then optimized accordingly.

**Questions:**

1. The introduction could be improved by adding a summary of contributions to provide clearer insight into the paper's impact and contribution.
2. Is it appropriate to set the same learning rate for different models (e.g. Mamba and Pythia) when comparing training stability? Whether it is possible that a better-tuned Transformer model could demonstrate enhanced stability and performance, suggesting that the comparison might benefit from adjustments in learning rate settings. Conducting an ablation study or sensitivity analysis on the learning rates for different models maybe can help.

---

> ### Author Response · Authors · 2024-11-23
> **Reply**
>
> We thank the reviewer for their helpful comments and suggestions.  We have added a summary of contributions to the introduction to provide clear insight into the paper's major contributions.
>
> # Is it appropriate to set the same learning rate for different models (e.g. Mamba and Pythia) when comparing training stability?
> We thank the reviewer for raising this question.   We first note that in Figure 1, the divergence is measured between a MPFT configuration and its full precision counterpart, e.g., BF16 LoRA versus FP32 LoRA or FP16 full fine-tuning versus FP32 full fine-tuning, with all other hyperparameters (e.g., batch size, LoRA dimension, learning rate, cosine annealing) set equivalently.  Thus, great care was taken to ensure that the source of divergence was due to the change in precision.
>
> With that said, we have run additional experiments to study the impact of hyperparameters on instruction tuning for Mamba and Pythia models (Appendix H of the updated submission).  Indeed, the results further demonstrate that Mamba models are more stable to variations in instruction tuning hyperparameters--e.g., the learning rate or LoRA dimension--compared to Pythia/Transformers.
>
> # MPFT and PEFT weaknesses
>
> We thank the reviewer for raising this point, we have also observed recent works adapting MPFT and PEFT for Mamba [1,2].  As newer works seek to leverage MPFT/PEFT for Mamba, we note this further warrants the need for rigorous theory to back up these decisions and justify why such decisions make sense.  We appreciate the suggestion to looks for MPFT and PEFT weaknesses.  Indeed, we believe our paper highlights a weakness associated in MPFT and PEFT, although not with Mamba models, but with Transformer LLMs.  We will be sure to point this out.
>
> Furthermore, given both our theoretical and extensive empirical results demonstrating the stability of Mamba LLMs under MPFT and PEFT--and the large instability of comparable Transformer LLMs--future work may leverage these results to fix this weakness for Transformer models.  Specifically, we will note that future work may explore extending the stability of Mamba models to Transfomers via hybrid models, which still routinely heavily quantize Transformer blocks while avoiding quantization of Mamba blocks [3].
>
> # References
> [1] Wang, Junxiong, et al. "The Mamba in the Llama: Distilling and Accelerating Hybrid Models." The Thirty-eighth Annual Conference on Neural Information Processing Systems.
>
> [2] Waleffe, Roger, et al. "An Empirical Study of Mamba-based Language Models." arXiv preprint arXiv:2406.07887 (2024).
>
> [3] Lieber, Opher, et al. "Jamba: A hybrid transformer-mamba language model." arXiv preprint arXiv:2403.19887 (2024).

---

> > ### Author Response · Authors · 2024-11-27
> > **Follow up**
> >
> > Dear Reviewer Pctm,
> >
> > We are following up to see whether the reviewer had a chance to review our reply and additional experiments addressing your concerns?  We note that the revised submission also includes additional experiments demonstrating the positive implications of Theorem 3, where regularization through LoRA in the MambaBlock near uniformly improves performance (Section 4.2).
> >
> > We also thank the reviewer for the earlier discussion wherein we found we could highlight our stability results as a weakness in MPFT and PEFT (two standard fine-tuning methods) for Transformer-based models, which can be solved using Mamba models given our main theoretical results.  We will be sure to incorporate this view into the manuscript.
> >
> > If the reviewer has any remaining concerns, please let us know.
> >
> > Best regards,
> >
> > Authors of Submission 9040

---

> > > ### Author Response · Authors · 2024-12-02
> > > **Following up as the discussion period closes**
> > >
> > > Dear Reviewer Pctm,
> > >
> > > As the discussion period closes, we wanted to follow up and double check whether the reviewer had any further questions/concerns given our replies.  To recap, we've added additional hyperparameter search experiments and addressed the reviewer's concern about learning rates in our earlier comments [here](https://openreview.net/forum?id=i9RTCC6whL&noteId=AQuiT90Mld).
> > >
> > > Additionally, we note that extensive experiments have been added further demonstrating Mamba SSMs' stability results (Appendix I) and demonstrating the generalization implications of Theorem 3 (Section 4.2).
> > >
> > > As previously noted, we thank the reviewer for their helpful feedback and aiding the authors articulate our stability results as a weakness of Transformer-based models, which may be solved by leveraging the extreme stability of Mamba models.
> > >
> > > Best,
> > >
> > > Authors of Submission 9040

---

### Author Response · Authors · 2024-11-24
**Updated results and manuscript**

We thank all reviewers for their feedback and time reviewing our paper.  We have updated the manuscript to reflect reviewer comments, including the following:
- Ablation results demonstrating the regularization implications of Theorem 3 and effects on improved performance generalization.
- Expanded divergence results, including additional fine-tuning datasets and benchmarks.  Agreeing with Reviewer's tWTf intuition, these extensive results further empirically confirm the paper's main theoretical results showing Mamba SSM's are stable learners, while also empirically demonstrating Transformer-based LLMs can be highly unstable when training using MPFT and PEFT.
- Writing changes: summary of major contributions was added to the introduction, shifting of the background section (which previously contained extra contextualization) to make way for more detailed empirical analysis in the main text, and reference fixes.  We thank the reviewers for these suggestions, we believe such changes have greatly improved the structure of the submission.
- Standard deviations to accompany mean divergence results, and raw performance numbers to accompany ICL results
- A study of the stability with respect to instruct tuning hyperparameters (Appendix H), which demonstrates Mamba models are far more stable to variations in such hyperparameters than Transformer LLMs.

Below, we also address general concerns from reviews.  We look forward to any additional feedback, please let us know if any points need further clarification or discussion.
# Theoretical results are shallow: Lemma 1 is trivial
We respectfully disagree.  The stability of recurrent neural models is not a given, with prior works promoting adjustment of the training schedule to combat unstable dynamics of such recurrent models [1, 2].  By theoretically showing Mamba SSMs are guaranteed to to be stable (Theorems 1 and 2), Mamba LLMs may directly use standard Transformer-based training frameworks (e.g., MPFT and PEFT) without adjusting for potential instabilities, as demonstrated at length in the paper.

Furthermore, calculating the lypanuv exponents of a recurrent model is nontrivial, i.e., we refer the reviewer to the proof of Lemma 1, which requires careful analysis of Mamba's state-space equations, variable activations, and variable domains.  Finally, we note that other reviews have highlighted the strength of the paper's theoretical results.

# No practical demonstration of Theorem 3
We have added new experiments (Section 4.2 of the updated manuscript) demonstrating the implications of Theorem 3 (i.e., parameter tying of time-varying parameters via LoRA's target layer).  The new results demonstrate that targeting the weight parameter described by Theorem 3 near uniformly results in better performance than only targeting other layers, while also showing that **only targeting this weight parameter outperforms targeting all linear layers using LoRA on a majority of tasks**.  The latter thus demonstrates the regularization implications of Theorem 3, further discussed at length in the updated submission.

# Recent Mamba MPFT works/MPFT and PEFT weaknesses for Mamba models
We thank the reviewer for raising this interesting point and their suggestion to find Mamba weaknesses for MPFT and PEFT.  As noted in the review, recent works [3, 4] have begun fine-tuning Mamba LLMs using mixed-precision.  We believe such decisions can only be strengthened when backed by strong theoretical results (such as the stability theorems featured in the submission).

We also note that **the paper highlights a weakness of the most widely used LLM architecture, i.e., that Transformer-based LLMs can be highly unstable when using some of the most common fine-tuning frameworks (e.g., MPFT and PEFT)**.  As a potential solution to such instability, the paper demonstrates MPFT and PEFT may be used with Mamba models without this weakness, and fine-tuned Mamba LLMs can demonstrate comparable emergent abilities (i.e., instruction tuned ICL) compared to Transformer-based LLMs.

# References
[1] Pascanu, R. "On the difficulty of training recurrent neural networks." arXiv preprint arXiv:1211.5063 (2013).

[2] Mikhaeil, Jonas, Zahra Monfared, and Daniel Durstewitz. "On the difficulty of learning chaotic dynamics with RNNs." Advances in Neural Information Processing Systems 35 (2022): 11297-11312.

[3] Wang, Junxiong, et al. "The Mamba in the Llama: Distilling and Accelerating Hybrid Models." The Thirty-eighth Annual Conference on Neural Information Processing Systems.

[4] Waleffe, Roger, et al. "An Empirical Study of Mamba-based Language Models." arXiv preprint arXiv:2406.07887 (2024).

---

### Author Response · Authors · 2024-12-03
**Comment from the authors**

Dear all,

The authors would like to thank Reviewers Pctm and tWTf for their constructive criticisms.  Based on this feedback, we have added a large number of experiments extensively validating Theorems 1 and 2 (spanning both different instruction tuning datasets and benchmarks), further highlighted the instability of Transformer-based LLMs for MPFT+PEFT in contrast to Mamba-1/2 LLMs, added hyperparameter search experiments demonstrating that Mamba LLMs are significantly more stable than Transformer LLMs for this paradigm, streamlined the writing of the paper, and added ablation experiments clearly demonstrating the generalization impact of Theorem 3.  Overall, these mark significant improvements to the paper, and we thank the reviewers again for their positive discussions.

To Reviewer jrwp: the reviewer has revealed their identity to the authors (without the authors' consent), requested additional experiments then admitted they never intended to positively adjust their review, repeatedly ignored author responses to their criticisms, and repeatedly stated incorrect claims which are not applicable to our results.  We thank the reviewer for stating Theorem 3 had no practical demonstration, and note these additional experiments were described in this [comment](https://openreview.net/forum?id=i9RTCC6whL&noteId=CsLrDeXS3Q).  We address your remaining criticisms below:

> Let’s be direct. I could simply choose not to respond, rate this submission poorly, and likely see it rejected.

Yes, thank you for not doing this.

> First, the stability analysis builds on well-known facts about RNN models, specifically that their decay terms are bounded by 1... they essentially restate the intuitive property that the exponential decay in linear RNNs...

This statement is incorrect.  The reviewer has stated they base their stability claims on [1].  [1] proves stability of linear *time-invariant* RNNs.  For such RNNs, [1] uses exponentiation of the time-invariant $A$ matrix during unrolling  (i.e., $\sum_j^{k-1} A^j$) to subsequently derive their stability results (Equation 4 of [[1]](https://arxiv.org/pdf/2303.06349).  In contrast, when the linear RNN is time-varying as in Mamba, i.e., $A_j$, unrolling does not maintain this exponentiation (instead, we have $\sum_{j=0}^{k-1} \prod_{i=j}^{k-1} A_i$, as described in Section 3.1 of [[2]](https://arxiv.org/pdf/2405.21060)).  The decay terms you mention are derived through eigenvalue analysis owing to the linear time-invariance of the RNNs in [1] (i.e., $\sum_j^{k-1} A^j = \sum_j^{k-1} P \Lambda^j P^{-1}$, where $A = P \Lambda P^{-1}$).  **This property does not hold for linear time varying Mamba SSMs, thus these decay terms do not exist for Mamba SSMs, and a stronger notion of stability is required, i.e., Lyapunov stability** (a cornerstone of stability analysis for linear dynamic systems [3,4]).  **Thus, our theoretical results are non-trivial and novel.**

> Second, I am not convinced that numerical stability is a significant issue to address here.

Thank you, we agree.  Indeed, this is the central thesis of the paper: **Mamba SSMs are naturally stable to numerical deviations.  Strong theory supports this, and extensive empirical evidence demonstrates this for MPFT/PEFT, in sharp contrast to Transformer LLMs.**

> To my knowledge, only matrix ( A ) needs to be stored in fp32, while all other linear projection parameters can safely use bf16—a method I’ve applied in practice without issues. I believe Mamba's official repository suggests is for the same reasons

This is incorrect, **the official Mamba repo states:**
> Our models were trained using PyTorch AMP for mixed precision. AMP keeps model parameters in float32 and casts to half precision when necessary... **SSMs are sensitive to their recurrent dynamics. If you are experiencing instabilities, as a first step please try a framework storing parameters in fp32**

Indeed, the last two sentences motivated us to theoretically characterize exactly how sensitive Mamba SSMs are to their recurrent dynamics.  Finally, the authors cannot verify the following: "a method I’ve applied in practice without issues."  Please provide citable, peer-reviewed evidence as the basis for criticisms.

# References
[1] Orvieto, Antonio, et al. "Resurrecting recurrent neural networks for long sequences." International Conference on Machine Learning. PMLR, 2023.  Arxiv link: https://arxiv.org/pdf/2303.06349

[2] Dao, Tri, and Albert Gu. "Transformers are SSMs: Generalized models and efficient algorithms through structured state space duality." arXiv preprint arXiv:2405.21060 (2024).

[3] Mikhaeil, Jonas, Zahra Monfared, and Daniel Durstewitz. "On the difficulty of learning chaotic dynamics with RNNs." Advances in Neural Information Processing Systems 35 (2022): 11297-11312.

[4] Yang, Lujie, et al. "Lyapunov-stable Neural Control for State and Output Feedback: A Novel Formulation." Forty-first International Conference on Machine Learning.

---

### Meta-Review · Area_Chair_HMj4 · 2024-12-25

**Metareview:**

Reviewer find the paper to have some interesting hypothesis on stability of Mamba models but expressed strong concerns over the choice of datasets and strength of theoretical results. Authors addressed some of these concerns by adding more experimental results, however reviewers are not fully convinced about the strength of the results. Overall I think this paper requires another round of rewriting to clearly present the hypothesis and the evidence. Hence I recommend rejection.

**Additional Comments On Reviewer Discussion:**

Main concern of reviewers were around strength of the results presented (both theoretical and empirical). Authors in their response included more experiments but couldn't fully convince the reviewers.

---

### Decision · Program_Chairs · 2025-01-22

Reject